# Synthesis of 1-Amino-3-oxo-2,7-naphthyridines *via* Smiles Rearrangement: A New Approach in the Field of Chemistry of Heterocyclic Compounds

**DOI:** 10.3390/ijms23115904

**Published:** 2022-05-25

**Authors:** Samvel N. Sirakanyan, Domenico Spinelli, Athina Geronikaki, Luca Zuppiroli, Riccardo Zuppiroli, Victor G. Kartsev, Elmira K. Hakobyan, Hasmik A. Yegoryan, Anush A. Hovakimyan

**Affiliations:** 1Scientific Technological Center of Organic and Pharmaceutical Chemistry of National Academy of Science of Republic of Armenia, Institute of Fine Organic Chemistry of A.L.Mnjoyan, Ave. Azatutyan 26, Yerevan 0014, Armenia; hakobyan.elmira@mail.ru (E.K.H.); hasmik.yegoryan@mail.ru (H.A.Y.); aaa.h.87@mail.ru (A.A.H.); 2Dipartimento di Chimica G. Ciamician, Alma Mater Studiorum-Università di Bologna, Via F. Selmi 2, 40126 Bologna, Italy; 3School of Pharmacy, Aristotle University of Thessaloniki, 54124 Thessaloniki, Greece; geronik@pharm.auth.gr; 4Department of Industrial Chemistry ‘Toso Montanari’, Alma Mater Studiorum-Università di Bologna, Viale del Risorgimento 4, 40136 Bologna, Italy; luca.zuppiroli@unibo.it (L.Z.); riccardo.zuppiroli@studio.unibo.it (R.Z.); 5InterBioScreen, 119019 Moscow, Russia; vkartsev@ibscreen.chg.ru

**Keywords:** 1-amino-3-oxo-2,7-naphthyridines, furo[2,3-*c*]-2,7-naphthyridines, 1,3-diamino-2,7-naphthyridines, Smiles rearrangement, alkylation, cyclization

## Abstract

In this paper we describe an efficient method for the synthesis of new heterocyclic systems: furo[2,3-*c*]-2,7-naphthyridines **6**, as well as a new method for the preparation of 1,3-diamino-2,7-naphthyridines **11**. For the first time, a Smiles rearrangement was carried out in the 2,7-naphthyridine series, thus gaining the opportunity to synthesize 1-amino-3-oxo-2,7-naphthyridines **4**, which are the starting compounds for obtaining furo[2,3-*c*]-2,7-naphthyridines. The cyclization of alkoxyacetamides **9** proceeds via two different processes: the expected formation of furo[2,3-*c*]-2,7-naphthyridines **10** and the ‘unexpected’ formation of 1,3-diamino-2,7-naphthyridines **11** (*via* a Smiles type rearrangement).

## 1. Introduction

2,7-Naphthyridine derivatives are interesting compounds in the field of heterocyclic chemistry due to some of their interesting biological activities, as evidenced by recent reviews [1,2,3]. Literature data shows that some 2,7-naphthyridine derivatives are potent and selective 3-phosphoinositide-dependent Kinase-1 [4], PDE5 [5], c-Kit/VEGFR-2 Kinase [6] inhibitors, while pyrazolo[3,4-*c*]-2,7-naphthyridines are bombesin receptor subtype-(3BRS-3) agonists [7].

Moreover, the literature has reported patents on the biological activity of 1-oxo-2,7-naphthyridines, showing that some derivatives of the latter are physiologically active compounds and provide anti-inflammatory and analgesic action and can be used as therapeutic agents for treatment of inflammatory immune diseases and chronic inflammations [8,9].

In our previous investigations on 2,7-naphthyridine derivatives, we have been able to synthesize some tricyclic heterocyclic systems, namely: isomeric 1,2,4-triazolo-2,7-naphthyridines **I** [10,11], pyrazolo[3,4-*c*]-2,7-naphthyridines **II** [12,13,14] and thieno[2,3-*c*]-2,7-naphthyridines **III** [15,16] as well as new heterocyclic systems based on these systems [13,14,15,16] (Figure 1).

Some of these investigations revealed that bicyclic 1,3-dihydroxy-2,7-naphthyridines can show antiarrhythmic activity [17], while tricyclic pyrazolo[3,4-*c*]-2,7-naphthyridines and triazolo[3,4-*a*]- as some triazolo[5,1-*a*]-2,7-naphthyridines displayed high neurotropic activity [10,12].

Furthermore, it is well known that fused derivatives of furo[2,3-*b*]pyridine show high biological activity [18,19,20,21,22,23,24]. Along this line we have previously synthesized several condensed furo[2,3-*b*]pyridines based on the cyclopenta[*c*]pyridines, 5,6,7,8-tetrahydroisoquinolines and pyrano[3,4-*c*]pyridines, obtaining compounds with neurotropic [25,26], antimicrobial [27], antitumor [28] and potent antiviral [29] activities.

Taking into account the abovementioned results, that are very interesting from both the chemical and the biological point of view, we think that it would be useful to combine these two heterocyclic systems into one molecule (Figure 1, compounds **IV**), as a ‘nice’ precondition for further interesting research. Until recently all our attempts to synthesize furo[2,3-*c*]-2,7-naphthyridines **IV** have failed and it is only in this study that we have been able to solve this problem by using the Smiles rearrangement. It should also be noted, that the furo[2,3-*c*]-2,7-naphthyridine **IV** system is currently not known in the literature.

## 2. Results and Discussion

For the synthesis of the targeted compounds we used, as a starting compound, 1,3-dichloro-7-isopropyl-5,6,7,8-tetrahydro-2,7-naphthyridine-4-carbonitrile (**1**) [16]. Its reaction with cyclic amines (pyrrolidine and azepane), in mild reaction conditions led to the formation of 1-amino-3-chloro-7-isopropyl-5,6,7,8-tetrahydro-2,7-naphthyridines **2a**,**b** [16]. In turn, compounds **2a**,**b** reacted with 2-mercaptoethanol giving the corresponding 1-amino-3-[(2-hydroxyethyl)thio]-7-isopropyl-5,6,7,8-tetrahydro-2,7-naphthyridine-4-carbonitriles **3a**,**b** (Figure 1) in high yields.

In a following step, compounds **3** under the action of sodium hydroxide in ethanol underwent a Smiles rearrangement [30,31,32] leading to the formation of 1-amino-7-isopropyl-3-oxo-2,3,5,6,7,8-hexahydro-2,7-naphthyridine-4-carbonitriles **4a**,**b** in quite high yields (Figure 1).

In compounds **4** a lactam–lactim (NH/OH) tautomerism is possible, via the proton-migration of a hydrogen atom between the two basic centers. The IR spectroscopic data strongly show that 1-amino-3-oxo-2,7-naphthyridines **4a**,**b** in the solid state exist only in the lactam **4** (**NH**) tautomeric form: showing carbonyl group absorptions at 1636–1638 cm^−1^, nitrile groups at 2208–2210 cm^−1^, and NH in the region 3222–3226 cm^−1^. Interestingly, in solution the situation changes, as confirmed by NMR spectra, where the presence of the proton of NH group at 10.53–10.86 ppm was observed (see Appendix A). Moreover, by further alkylation of 1-amino-3-oxo-2,7-naphthyridines **4a**,**b** in basic conditions, the corresponding *O*-alkylated derivatives were obtained in high yields (Schemes 3 and 4).

It should be mentioned that the literature reports similar *S*→*O* [33,34,35] and *S*→*N* [36,37,38,39] type Smiles rearrangement. The possible mechanism of formation of compounds **4** is presented in Figure 2. It is known that the cleavage product—thiirane—is formed in the reaction mixture and may polymerize under the basic conditions [33,34].

Then, for the synthesis of the targeted new heterocyclic systems, furo[2,3-*c*]-2,7-naphthyridines **7** (X = O), the obtained 3-oxo-2,7-naphthyridines **4a**,**b** were alkylated by ethyl chloroacetate in basic conditions. The synthesized *O*-alkylated compounds **5a**,**b** underwent cyclization in the presence of sodium ethoxide giving the fused furo[2,3-*c*]-2,7-naphthyridines **7a**,**b** (Figure 3).

A further confirmation of the structure of furo[2,3-*c*]-2,7-naphthyridines **7** is given by the fact that the *S*-alkylated compounds **6** gave the corresponding thieno[2,3-*c*]-2,7-naphthyridines **8**. In fact, in a first step 3-chloro-2,7-naphthyridines **2a,b** were alkylated with ethyl 2-mercaptoacetate and then the obtained intermediate *S*-alkylated derivatives **6a,b** were cyclized into the relevant thieno[2,3-*c*]-2,7-naphthyridines **8a,b** (Figure 3).

The structure of compounds **7** and **8** were confirmed by NMR, IR and MS spectroscopy and by elemental analysis (see Appendix A).

In addition, it was interesting to compare the physico-chemical properties of these two classes of compounds. Thus, furo[2,3-*c*]-2,7-naphthyridines **7** have a very higher solubility and lower melting point compared with thieno[2,3-*c*]-2,7-naphthyridines **8**. Moreover, compounds **7** and **8** showed different ^1^H NMR data: the singlet signal of the NH_2_ group in compounds **7a**,**b** were observed at 5.58 and 5.62 ppm, while in **8a**,**b** the same proton signal was shifted to a weaker field at 6.47 and 6.50 ppm. Signals of the remaining protons of these compounds differ slightly between themselves (Figure 2).

In order to increase the scope of furo[2,3-*c*]-2,7-naphthyridines, the obtained 3-oxo-2,7-naphthyridines **4** were alkylated by various other alkyl halides under basic conditions. The corresponding *O*-alkylated derivatives of 2,7-naphthyridine **9a**–**p** were obtained in high yields (75–89%, Figure 4, Table 1). In the ^1^H NMR spectra of compounds **9a**–**p** the singlet signals of the OCH_2_CO and NH groups were present at 4.69–4.92 ppm and 8.04–10.13 ppm, respectively (see Appendix A). The IR spectra also confirmed their structures, namely that they showed the absorption bands of nitrile group at 2201–2208 cm^−^^1^, of carbonyl group at 1662–1698 cm^−^^1^, and of NH group at 3166–3352 cm^−^^1^.

Moreover, to gain another way for obtaining the furo[2,3-*c*]-2,7-naphthyridines **7**, we refluxed the alkoxyacetamides **9** with sodium ethoxide in ethanol, following a route of reaction similar to that reported in Figure 3 for compounds **7**. Indeed, in this instance, different courses for the reaction were observed, depending on the structure of compounds **9** (Figure 4).

Thus, it was observed that amides deriving from cyclic (pyrrolidine and piperidine) **9a**,**b**,**h** or from aromatic amines **9c–g**,**i**–**l** gave only the expected aminoamides of furo[2,3-*c*]-2,7-naphthyridine **10a–l** in very good yields (70–83%). Therefore, it seems that in these cases only the activated CH_2_ group, which by the action of sodium ethoxide gave the relevant nucleophile CH**^−^**, was able to attack the nitrile group with formation of the condensed furan derivatives **10**, via an intramolecular nucleophilic addition (AN_i_).

In contrast, the amides deriving from non-aromatic primary amines **9m–o** (benzylamine or 2-furylmethylamine) furnished only the 1,3-diamino-2,7-naphthyridines **11a–c** again in good yields (71–74%). Thus, in these cases a Smiles type rearrangement occurred. Finally it must be remarked that in the instance of compound **9p** with primary amine [(1-methyl-2-phenylethyl)amine] no reaction occurred. This can be explained by the fact that, as is known, Smiles type rearrangements take place through the formation of a spiro-intermediate oxazolidinone ring (that is, a ‘Meisenheimer’ complex), the formation of which in this case is impossible due to the presence of the methyl near the NH group.

Such *O*→*N* Smiles type rearrangement was deeply investigated by us in the case of cycloalka[*c*]pyridine and pyrano[3,4-*c*]pyridine systems [40].

Concerning the structure of furo[2,3-*c*]-2,7-naphthyridines **10a**–**l** and of 1,3-diamino-2,7-naphthyridines **11a–c**, the first and most important information about their structure was given by IR spectra. In fact, the IR of the furo[2,3-*c*]-2,7-naphthyridines **10** showed the bands of the carbonyl group of the amide (1616–1654 cm^−^^1^) as well as of the NH_2_ group (3271–3488 cm^−^^1^), while the absorption bands of the nitrile group, characteristic of the initial compounds **9** and also confirmed by the ^13^C NMR spectrum, disappeared. Further, in the ^1^H NMR spectra of compounds **10a**–**l** the protons of the NH_2_ group at C-1 were observed at 5.61–5.83 ppm, whereas the signals of the OCH_2_CO group of starting **9a**–**l** were absent (see Appendix A).

The IR spectra of compounds **11** still showed the characteristic absorption bands of nitrile group at 2186–2199 cm^−^^1^, thus indicating that the cyclization process did not occur, as confirmed also by the ^13^C NMR spectrum. In the ^1^H NMR spectra of compounds **11** the protons of the NH group at 6.40–6.60 ppm were observed, while the signals of the OCH_2_CO group were absent. The ^13^C NMR spectra also confirmed their structure (see Appendix A).

The structure of compounds **10** and **11** was confirmed also by the MS spectroscopy.

Finally to gain a definitive confirmation of the structures of compounds **11**, their synthesis from the 3-chloro-2,7-naphthyridines **2a**,**b** was performed (Figure 4).

It must be remembered, that the observed Smiles type rearrangement represents a new and effective method for the synthesis of 1,3-diamino-2,7-naphthyridines **11**.

## 3. Materials and Methods

^1^H and ^13^C NMR spectra were recorded in DMSO*-**d**_6_*/CCl_4_ (1/3) solution (300 MHz for ^1^H and 75 MHz for ^13^C, respectively) on a Mercury 300VX spectrometer (Varian Inc., Palo Alto, CA, USA). Chemical shifts were reported as *δ* (parts per million) relative to TMS as internal standard. The IR spectra were recorded on a Nicolet Avatar 330-FT-IR spectrophotometer (Thermo Nicolet, Madison, CA, USA) in Vaseline, *ν_max_* in cm^–1^. MS spectra were recorded on Waters Q-Tof (Waters, Manchester, UK). Melting points were determined on an MP450 melting point apparatus. Elemental analyses were performed on an Elemental Analyzer Euro EA 3000. Compounds **1 [16]** and **2a [16]** have already been described.

### 3.1. Procedure for the Synthesis of Compound ***2b***

A mixture of compound **1** (2.70 g, 10 mmol), of hexamethyleneimine (1.24 mL, 11 mmol) and of triethylamine (1.53 mL, 11 mmol) in absolute ethanol (50 mL) was refluxed for 5 h. After cooling, water (50 mL) was added and the resulting crystals were filtered off, washed with water, dried, and recrystallized from ethanol.

1-Azepan-1-yl-3-chloro-7-isopropyl-5,6,7,8-tetrahydro-2,7-naphthyridine-4-carbonitrile (**2b**). Colorless solid; yield 87%, mp 137–139 °C; IR *ν*/cm^–1^: 2210 (C≡N). ^1^H NMR (300 MHz, DMSO*-**d_6_*/CCl_4_, 1/3): *δ* 1.09 (d, *J =* 6.5 Hz, 6H, CH(CH_3_)_2_), 1.56–1.65 (m, 4H, C_6_H_12_N), 1.77–1.86 (m, 4H, C_6_H_12_N), 2.70–2.78 (m, 2H, NCH_2_CH_2_), 2.82–2.92 (m, 3H, NCH_2_CH_2_, CH(CH_3_)_2_), 3.43 (br s, 2H, NCH_2_), 3.56–3.62 (m, 4H, N(CH_2_)_2_); ^13^C NMR (75 MHz, DMSO*-d_6_*/CCl_4_, 1/3): *δ* 18.02, 26.25, 27.86, 29.31, 44.00, 49.42, 50.96, 53.28, 97.62, 114.45, 116.65, 147.36, 150.09, 159.13. Anal. calcd. for C_18_H_25_ClN_4_: C 64.95; H 7.57; N 16.83%. Found: C 65.33; H 7.77; N 17.09%.

### 3.2. General Procedure for the Synthesis of Compounds ***3a,b***

To a stirred suspension of compound **2** (10 mmol) and potassium carbonate (2.76 g, 20 mmol) in absolute DMF (50 mL) the 2-mercaptoethanol (0.84 mL, 12 mmol) was added and the reaction mixture was stirred at 85–100 °C for 15 h. Cooling water was then added (50 mL). The resulting crystals were filtered off, washed with water, dried and recrystallized from ethanol.

3-[(2-Hydroxyethyl)thio]-7-isopropyl-1-pyrrolidin-1-yl-5,6,7,8-tetrahydro-2,7-naphthyridine-4-carbonitrile (**3a**). Yellow solid; yield 77%, mp 131–133 °C; IR *ν*/cm^–1^: 3143 (OH), 2202 (C≡N). ^1^H NMR (300 MHz, DMSO*-**d_6_*/CCl_4_, 1/3): *δ* 1.08 (d, *J =* 6.5 Hz, 6H, CH(CH_3_)_2_), 1.92–1.98 (m, 4H, C_4_H_8_N), 2.66–2.71 (m, 2H, NCH_2_CH_2_), 2.74–2.80 (m, 2H, NCH_2_CH_2_), 2.87 (sp, *J =* 6.5 Hz, 1H, CH(CH_3_)_2_), 3.22 (t, *J =* 6.8 Hz, 2H, SCH_2_), 3.54 (s, 2H, NCH_2_), 3.60 (q, *J =* 6.7 Hz, 2H, SCH_2_CH_2_), 3.61–3.67 (m, 4H, N(CH_2_)_2_), 4.55 (t, *J =* 5.7 Hz, 1H, OH); ^13^C NMR (75 MHz, DMSO*-**d_6_*/CCl_4_, 1/3): *δ* 18.03, 24.99, 28.75, 31.56, 44.03, 48.61, 49.38, 53.35, 60.28, 73.52, 112.39, 115.56, 147.53, 156.63, 157.25. Anal. calcd. for C_18_H_26_N_4_OS: C 62.39; H 7.56; N 16.17%. Found: C 62.72; H 7.73; N 16.39%.

1-Azepan-1-yl-3-[(2-hydroxyethyl)thio]-7-isopropyl-5,6,7,8-tetrahydro-2,7-naphthyridine-4-carbonitrile (**3b**). Light yellow solid; yield 74%, mp 151–153 °C; IR *ν*/cm^–1^: 3125 (OH), 2202 (C≡N). ^1^H NMR (300 MHz, DMSO*-d_6_*/CCl_4_, 1/3): *δ* 1.07 (d, *J =* 6.5 Hz, 6H, CH(CH_3_)_2_), 1.56–1.65 (m, 4H, C_6_H_12_N), 1.77–1.87 (m, 4H, C_6_H_12_N), 2.67–2.73 (m, 2H, NCH_2_CH_2_), 2.77–2.95 (m, 3H, NCH_2_CH_2_, CH(CH_3_)_2_), 3.23 (t, *J =* 6.8 Hz, 2H, SCH_2_), 3.39 (br s, 2H, NCH_2_), 3.56–3.64 (m, 6H, SCH_2_CH_2_, N(CH_2_)_2_), 4.57 (t, *J =* 5.6 Hz, 1H, OH); ^13^C NMR (75 MHz, DMSO*-d_6_*/CCl_4_, 1/3): *δ* 18.11, 26.44, 28.18, 28.91, 31.59, 44.33, 49.53, 50.72, 53.32, 60.23, 77.02, 113.66, 115.36, 148.42, 156.84, 158.98. Anal. calcd. for C_20_H_30_N_4_OS: C 64.13; H 8.07; N 14.96%. Found: C 64.48; H 8.26; N 15.20%.

### 3.3. General Procedure for the Synthesis of Compounds ***4a,b***

Aqueous solution of sodium hydroxide (50%, 8 g, 100 mmol) was added to a solution of compound **3** (10 mmol) in absolute ethanol (50 mL) and the mixture was refluxed for 15 h. After cooling, the water was added and the mixture was filtered off to remove white precipitate of the thiirane polymer. The filtrate was neutralized with HCl and the formed crystals of compound **4** were filtered off, washed with water and recrystallized from ethanol.

7-Isopropyl-3-oxo-1-pyrrolidin-1-yl-2,3,5,6,7,8-hexahydro-2,7-naphthyridine-4-carbonitrile (**4a**). Yellow solid; yield 79%, mp 222–223 °C; IR *ν*/cm^–1^: 3226 (NH), 2208 (C≡N), 1638 (C=O). ^1^H NMR (300 MHz, DMSO*-**d_6_*/CCl_4_, 1/3): *δ* 1.07 (d, *J =* 6.6 Hz, 6H, CH(CH_3_)_2_), 1.91–1.97 (m, 4H, 2CH_2_, C_6_H_12_N), 2.62–2.68 (m, 2H, NCH_2_CH_2_), 2.72–2.78 (m, 2H, NCH_2_CH_2_), 2.82 (sp, *J =* 6.6 Hz, 1H, CH(CH_3_)_2_), 3.46 (s, 2H, NCH_2_), 3.53–3.60 (m, 4H, N(CH_2_)_2_), 10.53 (br, 1H, NH); ^13^C NMR (75 MHz, DMSO*-**d_6_*/CCl_4_, 1/3): *δ* 18.07, 24.99, 29.19, 44.05, 48.62, 49.67, 53.37, 82.23, 105.16, 116.38, 151.57, 155.06, 161.14. Anal. calcd. for C_16_H_22_N_4_O: C 67.11; H 7.74; N 19.56%. Found: C 67.42; H 7.89; N 19.77%. ESI HRMS [C_16_H_22_O_1_N_4_+H^+^] Calculated: 287.1871. Found: 287.1873.

1-Azepan-1-yl-7-isopropyl-3-oxo-2,3,5,6,7,8-hexahydro-2,7-naphthyridine-4-carbonitrile (**4b**). Cream solid; yield 81%, mp 208–210 °C; IR *ν*/cm^–1^: 3222 (NH), 2210 (C≡N), 1636 (C=O). ^1^H NMR (300 MHz, DMSO*-d_6_*/CCl_4_, 1/3): *δ* 1.08 (d, *J =* 6.5 Hz, 6H, CH(CH_3_)_2_), 1.58–1.66 (m, 4H, C_6_H_12_N), 1.74–1.83 (m, 4H, C_6_H_12_N), 2.66–2.72 (m, 2H, NCH_2_CH_2_), 2.77–2.99 (m, 3H, NCH_2_CH_2_, CH(CH_3_)_2_), 3.34 (s, 2H, NCH_2_), 3.45–3.51 (m, 4H, N(CH_2_)_2_), 10.86 (br, 1H, NH). ^13^C NMR (75 MHz, DMSO*-d_6_*/CCl_4_, 1/3): *δ* 18.10, 26.48, 28.09, 29.08, 44.46, 48.98, 51.19, 53.34, 108.45, 115.78, 151.71, 158.23, 161.10. Anal. calcd. for C_18_H_26_N_4_O: C 68.76; H 8.33; N 17.82%. Found: C 69.13; H 8.51; N 18.07%. ESI HRMS C_18_H_26_O_1_N_4_+H^+^] Calculated: 315.2184. Found: 315.2186.

### 3.4. General Procedure for the Synthesis of Compounds ***5a,b*** and ***6a,b***

To a suspension of compound **4/2** (10 mmol) and potassium carbonate (2.76 g, 20 mmol) in absolute DMF (50 mL) ethyl chloroacetate (1.28 mL, 12 mmol) or ethyl 2-mercaptoacetate (1.32 mL, 12 mmol) was added dropwise under stirring. The reaction mixture was maintained at 75–80 °C for 3 h, then cooled to room temperature, and poured onto ice water. The resulting crystals were filtered off, washed with water, dried, and recrystallized from ethanol.

Ethyl [(4-cyano-7-isopropyl-1-pyrrolidin-1-yl-5,6,7,8-tetrahydro-2,7-naphthyridin-3-yl)oxy]acetate (**5a**). Light yellow solid; yield 73%, mp 98–100 °C; IR *ν*/cm^–1^: 2205 (C≡N), 1747 (C=O). ^1^H NMR (300 MHz, DMSO*-**d_6_*/CCl_4_, 1/3): *δ* 1.09 (d, *J =* 6.4 Hz, 6H, CH(CH_3_)_2_), 1.26 (t, *J =* 7.1 Hz, 3H, CH_2_CH_3_), 1.89–1.95 (m, 4H, C_4_H_8_N), 2.64–2.74 (m, 2H, NCH_2_CH_2_), 2.78–2.91 (m, 3H, NCH_2_CH_2_, CH(CH_3_)_2_), 3.52–3.60 (m, 6H, N(CH_2_)_2_, NCH_2_), 4.16 (q, *J =* 7.1 Hz, 2H, CH_2_CH_3_), 4.78 (s, 2H, OCH_2_); ^13^C NMR (75 MHz, DMSO*-**d_6_*/CCl_4_, 1/3): *δ* 13.76, 18.02, 24.90, 28.98, 43.92, 48.58, 49.17, 53.35, 59.83, 61.91, 81.35, 110.13, 114.84, 149.78, 156.21, 159.80, 167.50. Anal. calcd. for C_20_H_28_N_4_O_3_: C 64.49; H 7.58; N 15.04%. Found: C 64.83; H 7.74; N 15.28%.

Ethyl [(1-azepan-1-yl-4-cyano-7-isopropyl-5,6,7,8-tetrahydro-2,7-naphthyridin-3-yl)oxy]acetate (**5b**). Light yellow solid; yield 83%, mp 102–104 °C; IR *ν*/cm^–1^: 2204 (C≡N), 1745 (C=O). ^1^H NMR (300 MHz, DMSO*-d_6_*/CCl_4_, 1/3): *δ* 1.08 (d, *J =* 6.5 Hz, 6H, CH(CH_3_)_2_), 1.26 (t, *J =* 7.1 Hz, 3H, CH_2_CH_3_), 1.53–1.61 (m, 4H, C_6_H_12_N), 1.72–1.81 (m, 4H, C_6_H_12_N), 2.71 (t, *J* = 5.9 Hz, 2H, NCH_2_CH_2_), 2.79–2.89 (m, 3H, NCH_2_CH_2_, CH(CH_3_)_2_), 3.40 (s, 2H, NCH_2_), 3.49–3.54 (m, 4H, N(CH_2_)_2_), 4.15 (q, *J =* 7.1 Hz, 2H, CH_2_CH_3_), 4.81 (s, 2H, OCH_2_); ^13^C NMR (75 MHz, DMSO*-d_6_*/CCl_4_, 1/3): *δ* 13.71, 18.10, 26.22, 28.06, 29.16, 44.17, 49.56, 50.67, 53.32, 59.92, 61.77, 82.87, 111.35, 114.61, 150.70, 158.27, 159.41, 167.42. Anal. calcd. for C_22_H_32_N_4_O_3_: C 65.97; H 8.05; N 13.99%. Found: C 66.36; H 8.25; N 14.26%.

Ethyl [(4-cyano-7-isopropyl-1-pyrrolidin-1-yl-5,6,7,8-tetrahydro-2,7-naphthyridin-3-yl)thio]acetate (**6a**). Yellow solid; yield 76%, mp 114–116 °C; IR *ν*/cm^–1^: 2198 (C≡N), 1737 (C=O). ^1^H NMR (300 MHz, DMSO*-d_6_*/CCl_4_, 1/3): *δ* 1.08 (d, *J =* 6.5 Hz, 6H, CH(CH_3_)_2_), 1.26 (t, *J =* 7.1 Hz, 3H, CH_2_CH_3_), 1.92–1.98 (m, 4H, C_4_H_8_N), 2.70 (t, *J* = 5.9 Hz, 2H, NCH_2_CH_2_), 2.76–2.92 (m, 3H, NCH_2_CH_2_, CH(CH_3_)_2_), 3.55 (s, 2H, NCH_2_), 3.58–3.64 (m, 4H, N(CH_2_)_2_), 3.91 (s, 2H, SCH_2_), 4.12 (q, *J =* 7.1 Hz, 2H, CH_2_CH_3_); ^13^C NMR (75 MHz, DMSO*-d_6_*/CCl_4_, 1/3): *δ* 13.67, 18.00, 24.91, 28.74, 29.17, 31.11, 43.90, 48.60, 49.32, 49.59, 53.32, 60.36, 93.60, 112.88, 115.21, 147.61, 155.69, 156.62, 167.82. Anal. calcd. for C_20_H_28_N_4_O_2_S: C 61.83; H 7.26; N 14.42%. Found: C 62.15; H 7.43; N 14.65%.

Ethyl [(1-azepan-1-yl-4-cyano-7-isopropyl-5,6,7,8-tetrahydro-2,7-naphthyridin-3-yl)thio]acetate (**6b**). Light yellow solid; yield 74%, mp 103–105 °C; IR *ν*/cm^–1^: 2198 (C≡N), 1749 (C=O). ^1^H NMR (300 MHz, DMSO*-d_6_*/CCl_4_, 1/3): *δ* 1.08 (d, *J =* 6.5 Hz, 6H, CH(CH_3_)_2_), 1.26 (t, *J =* 7.1 Hz, 3H, CH_2_CH_3_), 1.58–1.63 (m, 4H, C_6_H_12_N), 1.76–1.85 (m, 4H, C_6_H_12_N), 2.71 (t, *J* = 5.9 Hz, 2H, NCH_2_CH_2_), 2.80–2.88 (m, 3H, NCH_2_CH_2_, CH(CH_3_)_2_), 3.40 (s, 2H, NCH_2_), 3.54–3.59 (m, 4H, N(CH_2_)_2_), 3.94 (s, 2H, SCH_2_), 4.12 (q, *J =* 7.1 Hz, 2H, CH_2_CH_3_); ^13^C NMR (75 MHz, DMSO*-d_6_*/CCl_4_, 1/3): *δ* 13.67, 18.08, 26.44, 28.11, 28.90, 30.97, 44.21, 49.53, 50.64, 53.30, 60.40, 60.43, 95.02, 114.28, 114.97, 148.60, 155.16, 159.07, 167.71. Anal. calcd. for C_22_H_32_N_4_O_2_S: C 63.43; H 7.74; N 13.45%. Found: C 63.83; H 7.95; N 13.73%.

### 3.5. General Procedure for the Synthesis of Compounds ***7a,b*** and ***8a,b***

To a solution of sodium ethoxide [0.25 g (11 mmol) in absolute ethanol (30 mL)] was added compound **5/6** (10 mmol). The mixture was refluxed for 1 h, cooled, and poured onto ice. The formed crystals were filtered off, washed with water, dried, and recrystallized from ethanol.

Ethyl 1-amino-7-isopropyl-5-pyrrolidin-1-yl-6,7,8,9-tetrahydrofuro[2,3-*c*]-2,7-naphthyridine-2-carboxylate (**7a**). Cream solid; yield 71%, mp 138–140 °C; IR *ν*/cm^–1^: 3503, 3381 (NH_2_), 1663 (C=O). ^1^H NMR (300 MHz, DMSO*-**d_6_*/CCl_4_, 1/3): *δ* 1.12 (d, *J =* 6.3 Hz, 6H, CH(CH_3_)_2_), 1.38 (t, *J =* 7.1 Hz, 3H, CH_2_CH_3_), 1.92–1.98 (m, 4H, C_4_H_8_N), 2.75–2.95 (m, 3H, NCH_2_CH_2_, CH(CH_3_)_2_), 3.15–3.22 (m, 2H, NCH_2_CH_2_), 3.49–3.56 (m, 4H, N(CH_2_)_2_), 3.57 (s, 2H, NCH_2_), 4.27 (q, *J =* 7.1 Hz, 2H, CH_2_CH_3_), 5.58 (s, 2H, NH_2_); ^13^C NMR (75 MHz, DMSO*-**d_6_*/CCl_4_, 1/3): *δ* 14.25, 17.85, 25.05, 26.50, 44.12, 48.81, 49.65, 53.90, 58.34, 102.45, 104.47, 119.98, 140.71, 140.96, 157.62, 157.73, 160.25. Anal. calcd. for C_20_H_28_N_4_O_3_: C 64.49; H 7.58; N 15.04%. Found: C 64.85; H 7.76; N 15.29%. ESI HRMS [C_20_H_28_O_3_N_4_+H^+^] Calculated: 373.2239. Found: 373.2241.

Ethyl 1-amino-5-azepan-1-yl-7-isopropyl-6,7,8,9-tetrahydrofuro[2,3-*c*]-2,7-naphthyridine-2-carboxylate (**7b**). Yellow solid; yield 69%, mp 147–149 °C; IR *ν*/cm^–1^: 3456, 3362 (NH_2_), 1676 (C=O). ^1^H NMR (300 MHz, DMSO*-d_6_*/CCl_4_, 1/3): *δ* 1.11 (d, *J =* 6.5 Hz, 6H, CH(CH_3_)_2_), 1.38 (t, *J =* 7.1 Hz, 3H, CH_2_CH_3_), 1.63–1.69 (m, 4H, C_6_H_12_N), 1.78–1.86 (m, 4H, C_6_H_12_N), 2.77 (t, *J* = 5.8 Hz, 2H, NCH_2_CH_2_), 2.87 (sp, *J =* 6.5 Hz, 1H, CH(CH_3_)_2_), 3.20 (t, *J* = 5.5 Hz, 2H, NCH_2_CH_2_), 3.44 (s, 2H, NCH_2_), 3.44–3.48 (m, 4H, N(CH_2_)_2_), 4.28 (q, *J =* 7.1 Hz, 2H, CH_2_CH_3_), 5.62 (s, 2H, NH_2_); ^13^C NMR (75 MHz, DMSO*-d_6_*/CCl_4_, 1/3): *δ* 14.24, 18.15, 26.37, 27.18, 28.35, 44.66, 49.61, 51.84, 53.41, 58.42, 103.95, 116.16, 120.36, 140.50, 141.89, 157.11, 160.32, 160.39. Anal. calcd. for C_22_H_32_N_4_O_3_: C 65.97; H 8.05; N 13.99%. Found: C 66.28; H 8.20; N 14.21%.

Ethyl 1-amino-7-isopropyl-5-pyrrolidin-1-yl-6,7,8,9-tetrahydrothieno[2,3-*c*]-2,7-naphthyridine-2-carboxylate (**8a**). Light yellow solid; yield 90%, mp 180–182 °C; IR *ν*/cm^–1^: 3441, 3335 (NH_2_), 1655 (C=O). ^1^H NMR (300 MHz, DMSO*-d_6_*/CCl_4_, 1/3): *δ* 1.11 (d, *J =* 6.5 Hz, 6H, CH(CH_3_)_2_), 1.36 (t, *J =* 7.1 Hz, 3H, CH_2_CH_3_), 1.92–1.98 (m, 4H, C_4_H_8_N), 2.75 (t, *J* = 5.8 Hz, 2H, NCH_2_CH_2_), 2.86 (sp, *J =* 6.5 Hz, 1H, CH(CH_3_)_2_), 3.26 (t, *J* = 5.9 Hz, 2H, NCH_2_CH_2_), 3.50–3.57 (m, 4H, N(CH_2_)_2_), 3.52 (s, 2H, NCH_2_), 4.24 (q, *J =* 7.1 Hz, 2H, CH_2_CH_3_), 6.47 (s, 2H, NH_2_); ^13^C NMR (75 MHz, DMSO*-d_6_*/CCl_4_, 1/3): *δ* 14.19, 18.17, 25.04, 27.63, 44.65, 49.57, 49.63, 53.35, 58.59, 90.91, 115.58, 115.98, 141.69, 150.50, 157.94, 157.98, 164.62. Anal. calcd. for C_20_H_28_N_4_O_2_S: C 61.83; H 7.26; N 14.42%. Found: C 62.18; H 7.43; N 14.66%. ESI HRMS [C_20_H_28_O_2_N_4_S_1_+H^+^] Calculated: 389.2011. Found: 389.2013.

Ethyl 1-amino-5-azepan-1-yl-7-isopropyl-6,7,8,9-tetrahydrothieno[2,3-*c*]-2,7-naphthyridine-2-carboxylate (**8b**). Light yellow solid; yield 89%, mp 162–164 °C; IR *ν*/cm^–1^: 3473, 3340 (NH_2_), 1659 (C=O). ^1^H NMR (300 MHz, DMSO*-d_6_*/CCl_4_, 1/3): *δ* 1.12 (d, *J =* 6.5 Hz, 6H, CH(CH_3_)_2_), 1.36 (t, *J =* 7.1 Hz, 3H, CH_2_CH_3_), 1.65–1.72 (m, 4H, C_6_H_12_N), 1.77–1.86 (m, 4H, C_6_H_12_N), 2.77 (t, *J* = 5.8 Hz, 2H, NCH_2_CH_2_), 2.86 (sp, *J =* 6.5 Hz, 1H, CH(CH_3_)_2_), 3.29 (t, *J* = 5.6 Hz, 2H, NCH_2_CH_2_), 3.44 (s, 2H, NCH_2_), 3.43–3.49 (m, 4H, N(CH_2_)_2_), 4.25 (q, *J =* 7.1 Hz, 2H, CH_2_CH_3_), 6.50 (s, 2H, NH_2_); ^13^C NMR (75 MHz, DMSO*-d_6_*/CCl_4_, 1/3): *δ* 14.17, 18.19, 26.44, 26.50, 27.63, 28.43, 28.48, 45.04, 49.63, 52.01, 53.32, 58.72, 91.93, 117.18, 117.79, 142.25, 150.25, 157.39, 160.98, 164.62. Anal. calcd. for C_22_H_32_N_4_O_2_S: C 63.43; H 7.74; N 13.45%. Found: C 63.82; H 7.94; N 13.73%.

### 3.6. General Procedure for the Synthesis of Compounds ***9a–p***

To a stirred suspension of compound **4** (1 mmol) and potassium carbonate (0.28 g, 2 mmol) in absolute DMF (25 mL) the corresponding alkyl chloride (1.2 mmol) was added. The reaction mixture was maintained at 75–80 °C for 3 h. Then the reaction mixture was cooled at room temperature, and water was added (50 mL). The resulting crystals were filtered off, washed with water, dried and recrystallized from ethanol.

7-Isopropyl-3-(2-oxo-2-pyrrolidin-1-ylethoxy)-1-pyrrolidin-1-yl-5,6,7,8-tetrahydro-2,7-naphthyridine-4-carbonitrile (**9a**). Milky solid; yield 75%, mp 178–180 °C; IR *ν*/cm^–1^: 2202 (C≡N), 1665 (C=O). ^1^H NMR (300 MHz, DMSO*-**d_6_*/CCl_4_, 1/3): *δ* 1.10 (d, *J =* 6.0 Hz, 6H, CH(CH_3_)_2_), 1.79–2.04 (m, 8H, 4CH_2_, C_4_H_8_N), 2.64–2.95 (m, 5H, NCH_2_CH_2_, CH(CH_3_)_2_), 3.37 (t, *J* = 6.8 Hz, 2H, NCH_2_), 3.50 (t, *J* = 6.8 Hz, 2H, NCH_2_), 3.53–3.60 (m, 6H, C_4_H_8_N, 8-CH_2_), 4.82 (s, 2H, OCH_2_); ^13^C NMR (75 MHz, DMSO*-**d_6_*/CCl_4_, 1/3): *δ* 17.97, 23.31, 24.91, 25.70, 28.90, 43.93, 44.72, 45.21, 48.48, 48.50, 49.08, 53.43, 63.41, 81.21, 115.11, 149.5, 156.24, 160.13, 164.79. Anal. calcd. for C_22_H_31_N_5_O_2_: C 66.47; H 7.86; N 17.62%. Found: C 66.80; H 8.02; N 17.87%.

7-Isopropyl-3-(2-oxo-2-piperidin-1-ylethoxy)-1-pyrrolidin-1-yl-5,6,7,8-tetrahydro-2,7-naphthyridine-4-carbonitrile (**9b**). Colorless solid; yield 80%, mp 164–166 °C; IR *ν*/cm^–1^: 2201 (C≡N), 1671 (C=O). ^1^H NMR (300 MHz, DMSO*-d_6_*/CCl_4_, 1/3): *δ* 1.09 (d, *J =* 6.5 Hz, 6H, CH(CH_3_)_2_), 1.46–1.71 (m, 6H, C_5_H_10_N), 1.90–1.95 (m, 4H, C_4_H_8_N), 2.66–2.91 (m, 5H, NCH_2_CH_2_, CH(CH_3_)_2_), 3.40–3.47 (m, 2H, NCH_2_), 3.54–3.61 (m, 8H, 2N(CH_2_)_2_, C_4_H_8_N, C_5_H_10_N), 4.92 (s, 2H, OCH_2_); ^13^C NMR (75 MHz, DMSO*-d_6_*/CCl_4_, 1/3): *δ* 18.02, 23.96, 24.91, 24.99, 25.02, 25.76, 28.92, 41.98, 42.02, 43.98, 44.88, 44.90, 48.54, 49.20, 53.39, 63.11, 81.33, 109.66, 115.12, 149.51, 156.27, 160.15, 164.45. Anal. calcd. for C_23_H_33_N_5_O_2_: C 67.12; H 8.08; N 17.02%. Found: C 67.49; H 8.27; N 17.29%.

2-[(4-Cyano-7-isopropyl-1-pyrrolidin-1-yl-5,6,7,8-tetrahydro-2,7-naphthyridin-3-yl)oxy]-*N*-phenylacetamide (**9c**). Cream solid; yield 78%, mp 227–229 °C; IR *ν*/cm^–1^: 3251 (NH), 2208 (C≡N), 1686 (C=O). ^1^H NMR (300 MHz, DMSO*-d_6_*/CCl_4_, 1/3): *δ* 1.07 (d, *J =* 6.5 Hz, 6H, CH(CH_3_)_2_), 1.79–1.88 (m, 4H, C_4_H_8_N), 2.68 (t, *J* = 5.7 Hz, 2H, NCH_2_CH_2_), 2.79–2.93 (m, 3H, NCH_2_CH_2_, CH(CH_3_)_2_), 3.51–3.58 (m, 6H, NCH_2_, N(CH_2_)_2_), 4.80 (s, 2H, OCH_2_), 6.95–7.02 (m, 1H, Ph), 7.19–7.26 (m, 2H, Ph), 7.54–7.59 (m, 2H, Ph), 9.68 (s, 1H, NH); ^13^C NMR (75 MHz, DMSO*-d_6_*/CCl_4_, 1/3): *δ* 18.03, 24.86, 28.98, 44.00, 48.56, 49.26, 53.37, 64.50, 81.23, 109.88, 115.35, 119.00, 122.60, 127.90, 138.37, 149.48, 156.37, 160.14, 165.62. Anal. calcd. for C_24_H_29_N_5_O_2_: C 68.71; H 6.97; N 16.69%. Found: C 69.03; H 7.12; N 16.92%.

2-[(4-Cyano-7-isopropyl-1-pyrrolidin-1-yl-5,6,7,8-tetrahydro-2,7-naphthyridin-3-yl)oxy]-*N*-(3-methylphenyl)acetamide (**9d**). Colorless solid; yield 83%, mp 222–224 °C; IR *ν*/cm^–1^: 3235, 3184 (NH), 2205 (C≡N), 1683 (C=O). ^1^H NMR (300 MHz, DMSO*-d_6_*/CCl_4_, 1/3): *δ* 1.08 (d, *J =* 6.5 Hz, 6H, CH(CH_3_)_2_), 1.81–1.88 (m, 4H, C_4_H_8_N), 2.31 (s, 3H, CH_3_), 2.67–2.72 (m, 2H, NCH_2_CH_2_), 2.81–2.93 (m, 3H, NCH_2_CH_2_, CH(CH_3_)_2_), 3.52–3.59 (m, 6H, NCH_2_, N(CH_2_)_2_), 4.79 (s, 2H, OCH_2_), 6.75–6.81 (m, 1H, C_6_H_4_), 7.06–7.15 (m, 1H, C_6_H_4_), 7.30–7.35 (m, 1H, C_6_H_4_), 7.39–7.41 (m, 1H, C_6_H_4_), 9.54 (s, 1H, NH); ^13^C NMR (75 MHz, DMSO*-d_6_*/CCl_4_, 1/3): *δ* 18.01, 20.97, 24.86, 28.92, 43.99, 48.53, 49.26, 53.41, 64.49, 81.25, 104.45, 115.32, 116.19, 119.61, 123.40, 127.75, 137.15, 138.21, 149.44, 156.36, 160.13, 165.50. Anal. calcd. for C_25_H_31_N_5_O_2_: C 69.26; H 7.21; N 16.15%. Found: C 69.62; H 7.39; N 16.40%.

*N*-(3-Chlorophenyl)-2-[(4-cyano-7-isopropyl-1-pyrrolidin-1-yl-5,6,7,8-tetrahydro-2,7-naphthyridin-3-yl)oxy]acetamide (**9e**). Cream solid; yield 76%, mp 206–208 °C; IR *ν*/cm^–1^: 3259, 3193 (NH), 2208 (C≡N), 1674 (C=O). ^1^H NMR (300 MHz, DMSO*-d_6_*/CCl_4_, 1/3): *δ* 1.07 (d, *J =* 6.5 Hz, 6H, CH(CH_3_)_2_), 1.80–1.88 (m, 4H, C_4_H_8_N), 2.68 (t, *J* = 5.8 Hz, 2H, NCH_2_CH_2_), 2.82 (t, *J* = 6.0 Hz, 2H, NCH_2_CH_2_), 2.84 (sp, *J =* 6.5 Hz, 1H, CH(CH_3_)_2_), 3.50–3.57 (m, 4H, N(CH_2_)_2_), 3.55 (s, 2H, NCH_2_), 4.79 (s, 2H, OCH_2_), 6.95–7.00 (m, 1H, C_6_H_4_), 7.20 (t, *J =* 8.1 Hz, 1H, C_6_H_4_), 7.46–7.50 (m, 1H, C_6_H_4_), 7.72 (t, *J =* 2.0 Hz, 1H, C_6_H_4_), 9.93 (s, 1H, NH); ^13^C NMR (75 MHz, DMSO*-d_6_*/CCl_4_, 1/3): *δ* 18.02, 24.85, 28.99, 43.99, 48.56, 49.26, 53.36, 64.53, 81.25, 109.95, 115.32, 117.12, 118.83, 122.39, 129.16, 133.13, 139.80, 149.51, 156.33, 160.07, 166.07. Anal. calcd. for C_24_H_28_ClN_5_O_2_: C 63.50; H 6.22; N 15.43%. Found: C 63.83; H 6.38; N 15.67%.

2-[(4-Cyano-7-isopropyl-1-pyrrolidin-1-yl-5,6,7,8-tetrahydro-2,7-naphthyridin-3-yl)oxy]-*N*-(3-methoxyphenyl)acetamide (**9f**). Cream solid; yield 82%, mp 195–197 °C; IR *ν*/cm^–1^: 3264, 3205 (NH), 2205 (C≡N), 1673 (C=O). ^1^H NMR (300 MHz, DMSO*-d_6_*/CCl_4_, 1/3): *δ* 1.07 (d, *J =* 6.5 Hz, 6H, CH(CH_3_)_2_), 1.80–1.89 (m, 4H, C_4_H_8_N), 2.68 (t, *J* = 5.9 Hz, 2H, NCH_2_CH_2_), 2.82 (t, *J* = 6.0 Hz, 2H, NCH_2_CH_2_), 2.84 (sp, *J =* 6.5 Hz, 1H, CH(CH_3_)_2_), 3.51–3.58 (m, 6H, NCH_2_, N(CH_2_)_2_), 3.76 (s, 3H, OCH_3_), 4.79 (s, 2H, OCH_2_), 6.51–6.55 (m, 1H, C_6_H_4_), 7.04–7.14 (m, 2H, C_6_H_4_), 7.28–7.30 (m, 1H, C_6_H_4_), 9.67 (s, 1H, NH); ^13^C NMR (75 MHz, DMSO*-d_6_*/CCl_4_, 1/3): *δ* 18.03, 24.87, 28.99, 44.00, 48.58, 49.26, 53.36, 54.35, 64.51, 81.23, 104.71, 108.47, 109.89, 111.20, 115.34, 128.54, 139.50, 149.48, 156.37, 159.20, 160.14, 165.66. Anal. calcd. for C_25_H_31_N_5_O_3_: C 66.79; H 6.95; N 15.58%. Found: C 67.17; H 7.14; N 15.85%.

2-[(4-Cyano-7-isopropyl-1-pyrrolidin-1-yl-5,6,7,8-tetrahydro-2,7-naphthyridin-3-yl)oxy]-*N*-(4-ethoxyphenyl)acetamide (**9g**). Colorless solid; yield 77%, mp 184–186 °C; IR *ν*/cm^–1^: 3238, 3188 (NH), 2203 (C≡N), 1680 (C=O). ^1^H NMR (300 MHz, DMSO*-d_6_*/CCl_4_, 1/3): *δ* 1.08 (d, *J =* 6.5 Hz, 6H, CH(CH_3_)_2_), 1.38 (t, *J* = 6.9 Hz, 3H, OCH_2_CH_3_), 1.82–1.89 (m, 4H, C_4_H_8_N), 2.55–2.72 (m, 2H, NCH_2_CH_2_), 2.79–2.89 (m, 3H, NCH_2_CH_2_, CH(CH_3_)_2_), 3.52–3.59 (m, 6H, NCH_2_, N(CH_2_)_2_), 3.97 (q, *J* = 6.9 Hz, 2H, OCH_2_CH_3_), 4.77 (s, 2H, OCH_2_), 6.72–6.78 (m, 2H, C_6_H_4_), 7.42–7.48 (m, 2H, C_6_H_4_), 9.46 (s, 1H, NH); ^13^C NMR (75 MHz, DMSO*-d_6_*/CCl_4_, 1/3): *δ* 14.38, 18.02, 24.88, 28.98, 43.99, 48.58, 49.27, 53.37, 62.53, 64.49, 81.30, 109.90, 113.68, 115.34, 120.44, 131.33, 149.47, 154.34, 156.37, 160.14, 165.10. Anal. calcd. for C_26_H_33_N_5_O_3_: C 67.36; H 7.18; N 15.11%. Found: C 67.68; H 7.33; N 15.34%.

1-Azepan-1-yl-7-isopropyl-3-(2-oxo-2-pyrrolidin-1-ylethoxy)-5,6,7,8-tetrahydro-2,7-naphthyridine-4-carbonitrile (**9h**). Light yellow solid; yield 75%, mp 173–175 °C; IR *ν*/cm^–1^: 2208 (C≡N), 1673 (C=O). ^1^H NMR (300 MHz, DMSO*-d_6_*/CCl_4_, 1/3): *δ* 1.08 (d, *J =* 6.5 Hz, 6H, CH(CH_3_)_2_), 1.53–1.60 (m, 4H, C_6_H_12_N), 1.71–1.80 (m, 4H, C_6_H_12_N), 1.80–1.89 (m, 2H, C_4_H_8_N), 1.94–2.04 (m, 2H, C_4_H_8_N), 2.70 (br t, *J* = 5.9 Hz, 2H, NCH_2_CH_2_), 2.79–2.89 (m, 3H, NCH_2_CH_2_, CH(CH_3_)_2_), 3.36 (t, *J* = 6.9 Hz, 2H, NCH_2_, C_4_H_8_N), 3.39 (s, 2H, NCH_2_), 3.44–3.55 (m, 6H, N(CH_2_)_2_, NCH_2_), 4.84 (s, 2H, OCH_2_); ^13^C NMR (75 MHz, DMSO*-d_6_*/CCl_4_, 1/3): *δ* 18.12, 23.30, 25.70, 26.18, 28.09, 29.16, 44.24, 44.47, 45.21, 49.56, 50.68, 53.33, 63.30, 82.82, 110.95, 114.94, 150.45, 158.22, 159.84, 164.59. Anal. calcd. for C_24_H_35_N_5_O_2_: C 67.73; H 8.29; N 16.46%. Found: C 68.13; H 8.50; N 16.75%.

2-[(1-Azepan-1-yl-4-cyano-7-isopropyl-5,6,7,8-tetrahydro-2,7-naphthyridin-3-yl)oxy]-*N*-(3-methylphenyl)acetamide (**9i**). Colorless solid; yield 81%, mp 183–185 °C; IR *ν*/cm^–1^: 3286, 3166 (NH), 2205 (C≡N), 1677 (C=O). ^1^H NMR (300 MHz, DMSO*-d_6_*/CCl_4_, 1/3): *δ* 1.08 (d, *J =* 6.6 Hz, 6H, CH(CH_3_)_2_), 1.46–1.54 (m, 4H, C_6_H_12_N), 1.67–1.77 (m, 4H, C_6_H_12_N), 2.32 (s, 3H, CH_3_), 2.71 (br t, *J* = 5.9 Hz, 2H, NCH_2_CH_2_), 2.79–2.90 (m, 3H, NCH_2_CH_2_, CH(CH_3_)_2_), 3.39 (m, 2H, NCH_2_), 3.47–3.57 (m, 4H, N(CH_2_)_2_), 4.80 (s, 2H, OCH_2_), 6.77–6.81 (m, 1H, C_6_H_4_), 7.06–7.12 (m, 1H, C_6_H_4_), 7.30–7.35 (m, 1H, C_6_H_4_), 7.38–7.41 (m, 1H, C_6_H_4_), 9.55 (s, 1H, NH); ^13^C NMR (75 MHz, DMSO*-d_6_*/CCl_4_, 1/3): *δ* 18.11, 20.97, 26.27, 28.04, 29.19, 44.27, 49.58, 50.69, 53.35, 64.37, 82.82, 111.09, 115.09, 116.14, 119.57, 123.39, 127.73, 137.11, 138.21, 150.40, 158.44, 159.78, 165.31. Anal. calcd. for C_27_H_35_N_5_O_2_: C 70.25; H 7.64; N 15.17%. Found: C 70.61; H 7.82; N 15.42%.

2-[(1-Azepan-1-yl-4-cyano-7-isopropyl-5,6,7,8-tetrahydro-2,7-naphthyridin-3-yl)oxy]-*N*-(4-methylphenyl)acetamide (**9j**). Colorless solid; yield 84%, mp 202–204 °C; IR *ν*/cm^–1^: 3245, 3187 (NH), 2208 (C≡N), 1683 (C=O). ^1^H NMR (300 MHz, DMSO*-d_6_*/CCl_4_, 1/3): *δ* 1.08 (d, *J =* 6.5 Hz, 6H, CH(CH_3_)_2_), 1.46–1.54 (m, 4H, C_6_H_12_N), 1.67–1.77 (m, 4H, C_6_H_12_N), 2.29 (s, 3H, CH_3_), 2.71 (t, *J* = 5.5 Hz, 2H, NCH_2_CH_2_), 2.79–2.91 (m, 3H, NCH_2_CH_2_, CH(CH_3_)_2_), 3.39 (s, 2H, NCH_2_), 3.47–3.53 (m, 4H, N(CH_2_)_2_), 4.80 (s, 2H, OCH_2_), 6.99–7.04 (m, 2H, C_6_H_4_), 7.40–7.46 (m, 2H, C_6_H_4_), 9.57 (s, 1H, NH); ^13^C NMR (75 MHz, DMSO*-d_6_*/CCl_4_, 1/3): *δ* 18.10, 20.27, 26.25, 28.03, 29.15, 29.18, 44.26, 49.54, 49.56, 49.59, 50.66, 53.35, 64.34, 82.77, 111.05, 115.10, 118.97, 128.36, 131.49, 135.80, 150.37, 158.42, 159.80, 165.15. Anal. calcd. for C_27_H_35_N_5_O_2_: C 70.25; H 7.64; N 15.17%. Found: C 70.64; H 7.85; N 15.44%.

2-[(1-Azepan-1-yl-4-cyano-7-isopropyl-5,6,7,8-tetrahydro-2,7-naphthyridin-3-yl)oxy]-*N*-(3-methoxyphenyl)acetamide (**9k**). Colorless solid; yield 89%, mp 170–172 °C; IR *ν*/cm^–1^: 3327, 3274 (NH), 2206 (C≡N), 1674 (C=O). ^1^H NMR (300 MHz, DMSO*-d_6_*/CCl_4_, 1/3): *δ* 1.07 (d, *J =* 6.5 Hz, 6H, CH(CH_3_)_2_), 1.45–1.54 (m, 4H, C_6_H_12_N), 1.67–1.76 (m, 4H, C_6_H_12_N), 2.71 (t, *J* = 5.8 Hz, 2H, NCH_2_CH_2_), 2.79–2.91 (m, 3H, NCH_2_CH_2_, CH(CH_3_)_2_), 3.39 (s, 2H, NCH_2_), 3.47–3.53 (m, 4H, N(CH_2_)_2_), 3.75 (s, 3H, OCH_3_), 4.81 (s, 2H, OCH_2_), 6.50–6.55 (m, 1H, 4-CH, C_6_H_4_), 7.03–7.13 (m, 2H, 5,6-CH, C_6_H_4_), 7.28–7.31 (m, 1H, 2-CH, C_6_H_4_), 9.68 (s, 1H, NH); ^13^C NMR (75 MHz, DMSO*-d_6_*/CCl_4_, 1/3): *δ* 18.11, 26.27, 28.04, 29.19, 44.26, 49.59, 50.67, 53.35, 54.32, 64.35, 82.76, 104.65, 108.46, 111.08, 111.14, 115.11, 128.51, 139.49, 150.40, 158.43, 159.17, 159.80, 165.47. Anal. calcd. for C_27_H_35_N_5_O_3_: C 67.90; H 7.39; N 14.66%. Found: C 68.23; H 7.55; N 14.90%.

*N*-(4-Acetylphenyl)-2-[(1-azepan-1-yl-4-cyano-7-isopropyl-5,6,7,8-tetrahydro-2,7-naphthyridin-3-yl)oxy]acetamide (**9l**). Light yellow solid; yield 76%, mp 198–200 °C; IR *ν*/cm^–1^: 3277, 3192 (NH), 2206 (C≡N), 1690, 1680 (C=O). ^1^H NMR (300 MHz, DMSO*-d_6_*/CCl_4_, 1/3): *δ* 1.07 (d, *J =* 6.5 Hz, 6H, CH(CH_3_)_2_), 1.44–1.51 (m, 4H, C_6_H_12_N), 1.65–1.74 (m, 4H, C_6_H_12_N), 2.51 (s, 3H, COCH_3_), 2.71 (t, *J* = 5.7 Hz, 2H, NCH_2_CH_2_), 2.78–2.91 (m, 3H, NCH_2_CH_2_, CH(CH_3_)_2_), 3.39 (s, 2H, NCH_2_), 3.45–3.51 (m, 4H, N(CH_2_)_2_), 4.85 (s, 2H, OCH_2_), 7.68–7.73 (m, 2H, C_6_H_4_), 7.82–7.87 (m, 2H, C_6_H_4_), 10.13 (s, 1H, NH); ^13^C NMR (75 MHz, DMSO*-d_6_*/CCl_4_, 1/3): *δ* 18.10, 25.64, 26.23, 28.02, 29.20, 44.25, 49.60, 50.63, 53.34, 64.36, 82.77, 111.13, 115.07, 118.12, 128.70, 131.50, 142.70, 150.47, 158.37, 159.74, 166.10, 194.55. Anal. calcd. for C_28_H_35_N_5_O_3_: C 68.69; H 7.21; N 14.30%. Found: C 69.04; H 7.39; N 14.56%.

*N*-Benzyl-2-[(4-cyano-7-isopropyl-1-pyrrolidin-1-yl-5,6,7,8-tetrahydro-2,7-naphthyridin-3-yl)oxy]acetamide (**9m**). Colorless solid; yield 80%, mp 173–175 °C; IR *ν*/cm^–1^: 3352 (NH), 2205 (C≡N), 1698 (C=O). ^1^H NMR (300 MHz, DMSO*-d_6_*/CCl_4_, 1/3): *δ* 1.09 (d, *J =* 6.6 Hz, 6H, CH(CH_3_)_2_), 1.81–1.90 (m, 4H, C_5_H_8_N), 2.69 (t, *J* = 5.6 Hz, 2H, NCH_2_CH_2_), 2.78–2.91 (m, 3H, NCH_2_CH_2_, CH(CH_3_)_2_), 3.48–3.55 (m, 4H, N(CH_2_)_2_), 3.57 (s, 2H, NCH_2_), 4.31 (d, *J =* 6.0 Hz, 2H, NHCH_2_), 4.70 (s, 2H, OCH_2_), 7.13–7.26 (m, 5H, Ph), 8.05 (t, *J =* 6.0 Hz, 1H, NH); ^13^C NMR (75 MHz, DMSO*-d_6_*/CCl_4_, 1/3): *δ* 18.03, 24.87, 28.99, 41.72, 43.99, 48.57, 49.19, 53.36, 64.23, 81.42, 109.84, 115.31, 126.03, 127.01, 127.48, 138.97, 149.47, 156.30, 160.03, 166.96. Anal. calcd. for C_25_H_31_N_5_O_2_: C 69.26; H 7.21; N 16.15%. Found: C 69.64; H 7.42; N 16.43%. ESI HRMS [C_25_H_31_O_2_N_5_+Na^+^] Calculated: 456.2375. Found: 456.2377.

2-[(1-Azepan-1-yl-4-cyano-7-isopropyl-5,6,7,8-tetrahydro-2,7-naphthyridin-3-yl)oxy]-*N*-benzylacetamide (**9n**). Colorless solid; yield 83%, mp 150–152 °C; IR *ν*/cm^–1^: 3197 (NH), 2206 (C≡N), 1675 (C=O). ^1^H NMR (300 MHz, DMSO*-d_6_*/CCl_4_, 1/3): *δ* 1.09 (d, *J =* 6.5 Hz, 6H, CH(CH_3_)_2_), 1.52–1.59 (m, 4H, C_6_H_12_N), 1.71–1.80 (m, 4H, C_6_H_12_N), 2.72 (t, *J* = 5.9 Hz, 2H, NCH_2_CH_2_), 2.81–2.90 (m, 3H, NCH_2_CH_2_, CH(CH_3_)_2_), 3.40 (s, 2H, NCH_2_), 3.46–3.51 (m, 4H, N(CH_2_)_2_), 4.31 (d, *J =* 5.9 Hz, 2H, CH_2_Ph), 4.72 (s, 2H, OCH_2_), 7.13–7.28 (m, 5H, Ph), 8.08 (t, *J* = 5.9 Hz, 1H, NH); ^13^C NMR (75 MHz, DMSO*-d_6_*/CCl_4_, 1/3): *δ* 18.14, 26.27, 28.09, 29.19, 41.75, 44.34, 49.52, 50.67, 53.36, 64.18, 82.89, 111.06, 115.10, 126.08, 126.99, 127.55, 138.89, 150.40, 158.44, 159.66, 166.82. Anal. calcd. for C_27_H_35_N_5_O_2_: C 70.25; H 7.64; N 15.17%. Found: C 70.65; H 7.86; N 15.47%. ESI HRMS [C_27_H_35_O_2_N_5_+Na^+^] Calculated: 484.2688. Found: 484.2689.

2-[(1-Azepan-1-yl-4-cyano-7-isopropyl-5,6,7,8-tetrahydro-2,7-naphthyridin-3-yl)oxy]-*N*-(2-furylmethyl)acetamide (**9o**). Light yellow solid; yield 77%, mp 154–156 °C; 152 ^o^C; IR *ν*/cm^–1^: 3288, 3189 (NH), 2206 (C≡N), 1678 (C=O). ^1^H NMR (300 MHz, DMSO*-d_6_*/CCl_4_, 1/3): *δ* 1.09 (d, *J =* 6.5 Hz, 6H, CH(CH_3_)_2_), 1.53–1.59 (m, 4H, C_6_H_12_N), 1.72–1.80 (m, 4H, C_6_H_12_N), 2.71 (t, *J* = 5.9 Hz, 2H, NCH_2_CH_2_), 2.80–2.90 (m, 3H, NCH_2_CH_2_, CH(CH_3_)_2_), 3.40 (s, 2H, NCH_2_), 3.48–3.53 (m, 4H, N(CH_2_)_2_), 4.30 (d, *J =* 5.7 Hz, 2H, NHCH_2_), 4.69 (s, 2H, OCH_2_), 6.14 (d, *J =* 3.1 Hz, 1H, 4-CH_fur_), 6.28 (dd, *J =* 3.1, 1.9 Hz, 1H, 3-CH_fur_), 7.36 (d, *J =* 1.5 Hz, 1H, 5-CH_fur_), 8.04 (t, *J* = 5.7 Hz, 1H, NH); ^13^C NMR (75 MHz, DMSO*-d_6_*/CCl_4_, 1/3): *δ* 18.12, 26.28, 28.09, 29.18, 35.03, 44.30, 49.54, 50.67, 53.34, 63.98, 82.84, 106.29, 109.73, 111.04, 115.06, 140.90, 150.38, 151.81, 158.43, 159.65, 166.73. Anal. calcd. for C_25_H_33_N_5_O_3_: C 66.50; H 7.37; N 15.51%. Found: C 66.82; H 7.54; N 15.75%. ESI HRMS [C_25_H_33_O_3_N_5_+Na^+^] Calculated: 474.2481. Found: 474.2482.

2-[(1-Azepan-1-yl-4-cyano-7-isopropyl-5,6,7,8-tetrahydro-2,7-naphthyridin-3-yl)oxy]-*N*-(1-methyl-2-phenylethyl)acetamide (**9p**). Colorless solid; yield 79%, mp 164–166 °C; IR *ν*/cm^–1^: 3306 (NH), 2207 (C≡N), 1662 (C=O). ^1^H NMR (300 MHz, DMSO*-d_6_*/CCl_4_, 1/3): *δ* 1.07 (d, *J =* 6.5 Hz, 6H, CH(CH_3_)_2_), 1.10 (d, *J =* 5.4 Hz, 3H, CHCH_3_), 1.53–1.59 (m, 4H, C_6_H_12_N), 1.73–1.81 (m, 4H, C_6_H_12_N), 2.55–2.92 (m, 7H, NCH_2_CH_2_, CH(CH_3_)_2_, CH_2_Ph), 3.40 (s, 2H, NCH_2_), 3.47–3.52 (m, 4H, N(CH_2_)_2_), 3.97–4.11 (m, 1H, NHCH), 4.54–4.66 (m, 2H, OCH_2_), 7.07–7.26 (m, 6H, NH, Ph); ^13^C NMR (75 MHz, DMSO*-d_6_*/CCl_4_, 1/3): *δ* 18.10, 19.42, 26.28, 28.09, 29.19, 41.76, 44.32, 45.60, 49.52, 50.72, 53.38, 64.09, 82.77, 111.14, 115.05, 125.39, 127.55, 128.60, 138.13, 150.38, 158.46, 159.59, 165.88. Anal. calcd. for C_29_H_39_N_5_O_2_: C 71.13; H 8.03; N 14.30%. Found: C 71.51; H 8.24; N 14.59%.

### 3.7. General Procedure for the Synthesis of Compounds ***10a–l***

To a solution of sodium ethoxide [0.05 g (2.2 mmol) of sodium in absolute ethanol (35 mL)] compound **5** (2 mmol) was added. The mixture was refluxed for 4 h, cooled, and poured onto water. The formed crystals were filtered off, washed with water, dried and recrystallized from ethanol.

7-Isopropyl-5-pyrrolidin-1-yl-2-(pyrrolidin-1-ylcarbonyl)-6,7,8,9-tetrahydrofuro[2,3-*c*]-2,7-naphthyridin-1-amine (**10a**). Milky solid; yield 77%, mp 250–252 °C; IR *ν*/cm^–1^: 3449, 3347 (NH_2_), 1619 (C=O). ^1^H NMR (300 MHz, DMSO*-**d_6_*/CCl_4_, 1/3): *δ* 1.13 (d, *J =* 6.3 Hz, 6H, CH(CH_3_)_2_), 1.91–1.98 (m, 8H, 4CH_2_, C_4_H_8_N), 2.72–2.93 (m, 3H, NCH_2_CH_2_, CH(CH_3_)_2_), 3.17–3.25 (m, 2H, NCH_2_CH_2_), 3.45–3.58 (m, 8H, 2(NCH_2_)_2_), 3.74 (br, 2H, NCH_2_), 5.64 (s, 2H, NH_2_); ^13^C NMR (75 MHz, DMSO*-**d_6_*/CCl_4_, 1/3): *δ* 18.04, 25.02, 26.90, 44.38, 46.08, 46.16, 49.17, 49.71, 53.57, 103.30, 113.98, 123.61, 138.57, 140.75, 156.44, 157.13, 160.32. Anal. calcd. for C_22_H_31_N_5_O_2_: C 66.47; H 7.84; N 17.62%. Found: C 66.83; H 8.06; N 17.89%. ESI HRMS [C_22_H_31_O_2_N_5_+H^+^] Calculated: 398.2556. Found: 398.2557.

7-Isopropyl-2-(piperidin-1-ylcarbonyl)-5-pyrrolidin-1-yl-6,7,8,9-tetrahydrofuro[2,3-*c*]-2,7-naphthyridin-1-amine (**10b**). Light yellow solid; yield 71%, mp 179–181 °C; IR *ν*/cm^–1^: 3427, 3306 (NH_2_), 1615 (C=O). ^1^H NMR (300 MHz, DMSO*-d_6_*/CCl_4_, 1/3): *δ* 1.11 (d, *J =* 6.5 Hz, 6H, CH(CH_3_)_2_), 1.60–1.74 (m, 6H, C_5_H_10_N), 1.91–1.97 (m, 4H, C_4_H_8_N), 2.76 (t, *J* = 5.9 Hz, 2H, NCH_2_CH_2_), 2.88 (sp, *J =* 6.5 Hz, 1H, CH(CH_3_)_2_), 3.19 (t, *J* = 5.8 Hz, 2H, NCH_2_CH_2_), 3.47–3.54 (m, 4H, N(CH_2_)_2_, C_5_H_10_N), 3.53 (s, 2H, NCH_2_), 3.74–3.79 (m, 4H, N(CH_2_)_2_, C_4_H_8_N), 5.67 (s, 2H, NH_2_); ^13^C NMR (75 MHz, DMSO*-d_6_*/CCl_4_, 1/3): *δ* 18.13, 24.49, 25.00, 25.89, 27.05, 44.45, 49.34, 49.38, 49.72, 53.43, 103.15, 114.34, 123.02, 139.74, 140.80, 156.06, 157.35, 160.70. Anal. calcd. for C_23_H_33_N_5_O_2_: C 67.12; H 8.08; N 17.02%. Found: C 67.45; H 8.24; N 17.27%.

1-Amino-7-isopropyl-*N*-phenyl-5-pyrrolidin-1-yl-6,7,8,9-tetrahydrofuro[2,3-*c*]-2,7-naphthyridine-2-carboxamide (**10c**). Cream solid; yield 73%, mp 196–198 °C; IR *ν*/cm^–1^: 3464, 3403, 3315 (NH, NH_2_), 1647 (C=O). ^1^H NMR (300 MHz, DMSO*-d_6_*/CCl_4_, 1/3): *δ* 1.12 (d, *J =* 6.5 Hz, 6H, CH(CH_3_)_2_), 1.91–1.99 (m, 4H, C_4_H_8_N), 2.77 (t, *J* = 5.8 Hz, 2H, NCH_2_CH_2_), 2.89 (sp, *J =* 6.5 Hz, 1H, CH(CH_3_)_2_), 3.21 (t, *J* = 5.5 Hz, 2H, NCH_2_CH_2_), 3.47–3.54 (m, 4H, N(CH_2_)_2_), 3.55 (s, 2H, NCH_2_), 5.67 (s, 2H, NH_2_), 6.92–6.98 (m, 1H, Ph), 7.18–7.25 (m, 2H, Ph), 7.78–7.83 (m, 2H, Ph), 9.25 (s, 1H, NH); ^13^C NMR (75 MHz, DMSO*-d_6_*/CCl_4_, 1/3): *δ* 18.16, 25.07, 27.10, 44.41, 49.38, 49.78, 53.51, 103.72, 114.28, 119.53, 121.88, 122.64, 127.66, 138.44, 139.14, 141.34, 156.65, 157.29, 159.25. Anal. calcd. for C_24_H_29_N_5_O_2_: C 68.71; H 6.97; N 16.69%. Found: C 69.10; H 7.17; N 16.97%. ESI HRMS [C_24_H_29_O_2_N_5_+H^+^] Calculeted: 420.2399. Found: 420.2400.

1-Amino-7-isopropyl-*N*-(3-methylphenyl)-5-pyrrolidin-1-yl-6,7,8,9-tetrahydrofuro[2,3-*c*]-2,7-naphthyridine-2-carboxamide (**10d**). Colorless solid; yield 70%, mp 108–110 °C; IR *ν*/cm^–1^: 3457, 3413, 3332 (NH, NH_2_), 1637 (C=O). ^1^H NMR (300 MHz, DMSO*-d_6_*/CCl_4_, 1/3): *δ* 1.12 (d, *J =* 6.5 Hz, 6H, CH(CH_3_)_2_), 1.90–1.97 (m, 4H, C_4_H_8_N), 2.32 (s, 3H, CH_3_), 2.77 (t, *J* = 5.4 Hz, 2H, NCH_2_CH_2_), 2.91 (sp, *J =* 6.5 Hz, 1H, CH(CH_3_)_2_), 3.20 (t, *J* = 5.5 Hz, 2H, NCH_2_CH_2_), 3.48–3.52 (m, 4H, N(CH_2_)_2_), 3.58 (s, 2H, NCH_2_), 5.64 (s, 2H, NH_2_), 6.74–6.80 (m, 1H, C_6_H_4_), 7.04–7.13 (m, 1H, C_6_H_4_), 7.54–7.59 (m, 1H, C_6_H_4_), 7.63–7.68 (m, 1H, C_6_H_4_), 9.10 (s, 1H, NH); ^13^C NMR (75 MHz, DMSO*-d_6_*/CCl_4_, 1/3): *δ* 18.15, 21.08, 25.04, 27.11, 44.40, 49.38, 49.76, 53.47, 103.74, 114.32, 116.66, 120.09, 122.64, 122.68, 127.52, 136.77, 138.33, 138.93, 141.31, 156.58, 157.26, 159.16. Anal. calcd. for C_25_H_31_N_5_O_2_: C 69.26; H 7.21; N 16.15%. Found: C 69.57; H 7.38; N 16.39%. ESI HRMS [C_25_H_31_O_2_N_5_+H^+^] Calculated: 434.2556. Found: 434.2557.

1-Amino-*N*-(3-chlorophenyl)-7-isopropyl-5-pyrrolidin-1-yl-6,7,8,9-tetrahydrofuro[2,3-*c*]-2,7-naphthyridine-2-carboxamide (**10e**). Light yellow solid; yield 83%, mp 202–204 °C; IR *ν*/cm^–1^: 3383, 3271 (NH, NH_2_), 1647 (C=O). ^1^H NMR (300 MHz, DMSO*-d_6_*/CCl_4_, 1/3): *δ* 1.12 (d, *J =* 6.5 Hz, 6H, CH(CH_3_)_2_), 1.91–2.00 (m, 4H, C_4_H_8_N), 2.77 (t, *J* = 5.7 Hz, 2H, NCH_2_CH_2_), 2.88 (sp, *J =* 6.5 Hz, 1H, CH(CH_3_)_2_), 3.21 (t, *J* = 5.7 Hz, 2H, NCH_2_CH_2_), 3.47–3.54 (m, 4H, N(CH_2_)_2_), 3.56 (s, 2H, NCH_2_), 5.75 (s, 2H, NH_2_), 6.90–6.94 (m, 1H, C_6_H_4_), 7.18 (t, *J* = 8.1 Hz, 1H, C_6_H_4_), 7.69–7.74 (m, 1H, C_6_H_4_), 8.05 (t, *J* = 2.0 Hz, 1H, C_6_H_4_), 9.57 (s, 1H, NH); ^13^C NMR (75 MHz, DMSO*-d_6_*/CCl_4_, 1/3): *δ* 18.13, 25.05, 27.09, 44.36, 49.35, 49.77, 53.48, 103.51, 114.26, 117.59, 119.20, 121.43, 122.29, 128.75, 132.80, 139.05, 140.73, 141.41, 156.79, 157.38, 159.28. Anal. calcd. for C_24_H_28_ClN_5_O_2_: C 63.50; H 6.22; N 15.43%. Found: C 63.87; H 6.41; N 15.70%. ESI HRMS [C_24_H_28_O_2_N_5_Cl_1_+H^+^] Calculeted: 454.2009. Found: 454.2011.

1-Amino-7-isopropyl-*N*-(3-methoxyphenyl)-5-pyrrolidin-1-yl-6,7,8,9-tetrahydrofuro[2,3-*c*]-2,7-naphthyridine-2-carboxamide (**10f**). Cream solid; yield 76%, mp 189–191 °C; IR *ν*/cm^–1^: 3374, 3276 (NH, NH_2_), 1651 (C=O). ^1^H NMR (300 MHz, DMSO*-d_6_*/CCl_4_, 1/3): *δ* 1.12 (d, *J =* 6.5 Hz, 6H, CH(CH_3_)_2_), 1.91–1.99 (m, 4H, C_4_H_8_N), 2.77 (t, *J* = 5.9 Hz, 2H, NCH_2_CH_2_), 2.89 (sp, *J =* 6.5 Hz, 1H, CH(CH_3_)_2_), 3.21 (t, *J* = 5.7 Hz, 2H, NCH_2_CH_2_), 3.47–3.54 (m, 4H, N(CH_2_)_2_), 3.55 (s, 2H, NCH_2_), 3.78 (s, 3H, OCH_3_), 5.69 (s, 2H, NH_2_), 6.47–6.52 (m, 1H, C_6_H_4_), 7.09 (t, *J* = 8.1 Hz, 1H, C_6_H_4_), 7.34–7.38 (m, 1H, C_6_H_4_), 7.55 (t, *J* = 2.2 Hz, 1H, C_6_H_4_), 9.21 (s, 1H, NH); ^13^C NMR (75 MHz, DMSO*-d_6_*/CCl_4_, 1/3): *δ* 18.15, 25.05, 27.11, 44.40, 49.38, 49.76, 53.48, 54.33, 103.71, 104.91, 108.01, 111.72, 114.31, 122.58, 128.23, 138.49, 140.29, 141.35, 156.62, 157.29, 159.04, 159.22. Anal. calcd. for C_25_H_31_N_5_O_3_: C 66.79; H 6.95; N 15.58%. Found: C 67.13; H 7.11; N 15.83%. ESI HRMS [C_25_H_31_O_3_N_5_+H^+^] Calculated: 450.2505. Found: 450.2507.

1-Amino-*N*-(4-ethoxyphenyl)-7-isopropyl-5-pyrrolidin-1-yl-6,7,8,9-tetrahydrofuro[2,3-*c*]-2,7-naphthyridine-2-carboxamide (**10g**). Light yellow solid; yield 70%, mp 187–189 °C; IR *ν*/cm^–1^: 3405, 3316 (NH, NH_2_), 1643 (C=O). ^1^H NMR (300 MHz, DMSO*-d_6_*/CCl_4_, 1/3): *δ* 1.13 (d, *J =* 6.5 Hz, 6H, CH(CH_3_)_2_), 1.39 (t, *J* = 7.0 Hz, 3H, OCH_2_CH_3_), 1.92–1.98 (m, 4H, C_4_H_8_N), 2.79 (br t, *J* = 5.6 Hz, 2H, NCH_2_CH_2_), 2.84–2.96 (m, 1H, CH(CH_3_)_2_), 3.22 (t, *J* = 5.6 Hz, 2H, NCH_2_CH_2_), 3.47–3.53 (m, 4H, N(CH_2_)_2_), 3.57 (s, 2H, NCH_2_), 3.99 (q, *J* = 7.0 Hz, 2H, OCH_2_CH_3_), 5.61 (s, 2H, NH_2_), 6.72–6.77 (m, 2H, C_6_H_4_), 7.64–7.70 (m, 2H, C_6_H_4_), 9.15 (s, 1H, NH); ^13^C NMR (75 MHz, DMSO*-d_6_*/CCl_4_, 1/3): *δ* 14.50, 18.08, 25.07, 26.91, 44.37, 49.24, 49.77, 53.62, 62.59, 103.85, 113.53, 113.94, 120.95, 122.87, 132.13, 137.86, 141.13, 153.92, 156.57, 157.13, 159.01. Anal. calcd. for C_26_H_33_N_5_O_3_: C 67.36; H 7.18; N 15.11%. Found: C 67.76; H 7.39; N 15.40%. ESI HRMS [C_26_H_33_O_3_N_5_+H^+^] Calculated: 464.2661. Found: 464.2663.

5-Azepan-1-yl-7-isopropyl-2-(pyrrolidin-1-ylcarbonyl)-6,7,8,9-tetrahydrofuro[2,3-*c*]-2,7-naphthyridin-1-amine (**10h**). Yellow solid; yield 75%, mp 198–200 °C; IR *ν*/cm^–1^: 3446, 3349 (NH_2_), 1616 (C=O). ^1^H NMR (300 MHz, DMSO*-d_6_*/CCl_4_, 1/3): *δ* 1.12 (d, *J =* 6.5 Hz, 6H, CH(CH_3_)_2_), 1.66–1.72 (m, 4H, 2CH_2_), 1.77–1.85 (m, 4H, 2CH_2_), 1.90–1.99 (m, 4H, 2CH_2_), 2.77 (t, *J* = 5.8 Hz, 2H, NCH_2_CH_2_), 2.87 (sp, *J =* 6.5 Hz, 1H, CH(CH_3_)_2_), 3.22 (t, *J* = 5.8 Hz, 2H, NCH_2_CH_2_), 3.38–3.43 (m, 4H, N(CH_2_)_2_, C_6_H_12_N), 3.48 (s, 2H, NCH_2_), 3.54–3.97 (m, 4H, N(CH_2_)_2_, C_4_H_8_N), 5.65 (s, 2H, NH_2_); ^13^C NMR (75 MHz, DMSO*-d_6_*/CCl_4_, 1/3): *δ* 18.15, 26.45, 27.08, 28.46, 44.84, 46.12, 49.36, 52.29, 53.40, 104.92, 116.78, 124.16, 138.26, 141.30, 156.03, 159.94, 160.32, 165.48. Anal. calcd. for C_24_H_35_N_5_O_2_: C 67.73; H 8.29; N 16.46%. Found: C 68.08; H 8.47; N 16.72%. ESI HRMS [C_24_H_35_O_2_N_5_+H^+^] Calculeted: 426.2869. Found: 426.2871.

1-Amino-5-azepan-1-yl-7-isopropyl-*N*-(3-methylphenyl)-6,7,8,9-tetrahydrofuro[2,3-*c*]-2,7-naphthyridine-2-carboxamide (**10i**). Colorless solid; yield 80%, mp 91–93 °C; IR *ν*/cm^–1^: 3398, 3304 (NH, NH_2_), 1654 (C=O). ^1^H NMR (300 MHz, DMSO*-d_6_*/CCl_4_, 1/3): *δ* 1.13 (d, *J =* 6.5 Hz, 6H, CH(CH_3_)_2_), 1.66–1.72 (m, 4H, C_6_H_12_N), 1.78–1.87 (m, 4H, C_6_H_12_N), 2.34 (s, 3H, CH_3_), 2.80 (br t, *J* = 5.6 Hz, 2H, NCH_2_CH_2_), 2.83–2.95 (m, 1H, CH(CH_3_)_2_), 3.25 (t, *J* = 5.5 Hz, 2H, NCH_2_CH_2_), 3.41–3.46 (m, 4H, N(CH_2_)_2_), 3.50 (s, 2H, NCH_2_), 5.69 (s, 2H, NH_2_), 6.75–6.79 (m, 1H, C_6_H_4_), 7.07–7.13 (m, 1H, C_6_H_4_), 7.52–7.57 (m, 1H, C_6_H_4_), 7.64–7.66 (m, 1H, C_6_H_4_), 9.10 (s, 1H, NH); ^13^C NMR (75 MHz, DMSO*-d_6_*/CCl_4_, 1/3): *δ* 18.14, 21.06, 26.41, 27.07, 28.42, 44.76, 49.44, 52.09, 53.45, 105.33, 116.67, 120.11, 122.79, 123.20, 127.53, 136.78, 137.97, 138.79, 141.73, 156.16, 159.15, 159.96. Anal. calcd. for C_27_H_35_N_5_O_2_: C 70.25; H 7.64; N 15.17%. Found: C 70.63; H 7.84; N 15.44%. ESI HRMS [C_27_H_35_O_2_N_5_+H^+^] Calculated: 462.2869. Found: 462.2870.

1-Amino-5-azepan-1-yl-7-isopropyl-*N*-(4-methylphenyl)-6,7,8,9-tetrahydrofuro[2,3-*c*]-2,7-naphthyridine-2-carboxamide (**10j**). Yellow solid; yield 77%, mp 204–206 °C; IR *ν*/cm^–1^: 3485, 3397, 3316 (NH, NH_2_), 1647 (C=O). ^1^H NMR (300 MHz, DMSO*-d_6_*/CCl_4_, 1/3): *δ* 1.13 (d, *J =* 6.5 Hz, 6H, CH(CH_3_)_2_), 1.65–1.73 (m, 4H, C_6_H_12_N), 1.78–1.87 (m, 4H, C_6_H_12_N), 2.31 (s, 3H, CH_3_), 2.79 (t, *J* = 5.7 Hz, 2H, NCH_2_CH_2_), 2.84–2.94 (m, 1H, CH(CH_3_)_2_), 3.24 (t, *J* = 5.7 Hz, 2H, NCH_2_CH_2_), 3.40–3.46 (m, 4H, N(CH_2_)_2_), 3.49 (s, 2H, NCH_2_), 5.67 (s, 2H, NH_2_), 7.00–7.05 (m, 2H, C_6_H_4_), 7.63–7.68 (m, 2H, C_6_H_4_), 9.15 (s, 1H, NH). ^13^C NMR (75 MHz, DMSO*-d_6_*/CCl_4_, 1/3): *δ* 18.15, 20.30, 26.41, 27.08, 28.42, 44.77, 49.46, 52.09, 53.44, 105.37, 116.60, 119.55, 123.28, 128.18, 130.81, 136.39, 137.79, 141.72, 156.13, 159.10, 159.91. Anal. calcd. for C_27_H_35_N_5_O_2_: C 70.25; H 7.64; N 15.17%. Found: C 70.56; H 7.79; N 15.40%. ESI HRMS [C_27_H_35_O_2_N_5_+H^+^] Calculated: 462.2869. Found: 462.2870.

1-Amino-5-azepan-1-yl-7-isopropyl-*N*-(3-methoxyphenyl)-6,7,8,9-tetrahydrofuro[2,3-*c*]-2,7-naphthyridine-2-carboxamide (**10k**). Colorless solid; yield 72%, mp 210–212 °C; IR *ν*/cm^–1^: 3488, 3415, 3334 (NH, NH_2_), 1650 (C=O). ^1^H NMR (300 MHz, DMSO*-d_6_*/CCl_4_, 1/3): *δ* 1.13 (d, *J =* 6.5 Hz, 6H, CH(CH_3_)_2_), 1.66–1.72 (m, 4H, C_6_H_12_N), 1.79–1.87 (m, 4H, C_6_H_12_N), 2.79 (t, *J* = 5.9 Hz, 2H, NCH_2_CH_2_), 2.89 (sp, *J =* 6.5 Hz, 1H, CH(CH_3_)_2_), 3.24 (t, *J* = 5.7 Hz, 2H, NCH_2_CH_2_), 3.41–3.46 (m, 4H, N(CH_2_)_2_), 3.49 (s, 2H, NCH_2_), 3.79 (s, 3H, OCH_3_), 5.71 (s, 2H, NH_2_), 6.48–6.53 (m, 1H, C_6_H_4_), 7.07–7.13 (m, 1H, C_6_H_4_), 7.33–7.38 (m, 1H, C_6_H_4_), 7.54–7.56 (m, 1H, C_6_H_4_), 9.22 (s, 1H, NH); ^13^C NMR (75 MHz, DMSO*-d_6_*/CCl_4_, 1/3): *δ* 18.15, 26.40, 27.12, 28.40, 44.77, 49.48, 52.08, 53.41, 54.30, 104.93, 105.28, 108.11, 111.73, 116.62, 123.11, 128.23, 138.14, 140.15, 141.80, 156.18, 159.04, 159.21, 160.0. Anal. calcd. for C_27_H_35_N_5_O_3_: C 67.90; H 7.39; N 14.66%. Found: C 68.29; H 7.60; N 14.94%. ESI HRMS [C_27_H_35_O_3_N_5_+H^+^] Calculated: 478.2818. Found: 478.2816.

*N*-(4-Acetylphenyl)-1-amino-5-azepan-1-yl-7-isopropyl-6,7,8,9-tetrahydrofuro[2,3-*c*]-2,7-naphthyridine-2-carboxamide (**10l**). Light yellow solid; yield 81%, mp 116–118 °C; IR *ν*/cm^–1^: 3399, 3312 (NH, NH_2_), 1673, 1648 (C=O). ^1^H NMR (300 MHz, DMSO*-d_6_*/CCl_4_, 1/3): *δ* 1.13 (d, *J =* 6.5 Hz, 6H, CH(CH_3_)_2_), 1.66–1.72 (m, 4H, C_6_H_12_N), 1.79–1.87 (m, 4H, C_6_H_12_N), 2.51 (s, 3H, COCH_3_), 2.80 (br t, *J* = 5.6 Hz, 2H, NCH_2_CH_2_), 2.84–2.94 (m, 1H, CH(CH_3_)_2_), 3.25 (t, *J* = 5.7 Hz, 2H, NCH_2_CH_2_), 3.42–3.47 (m, 4H, N(CH_2_)_2_), 3.50 (s, 2H, NCH_2_), 5.83 (s, 2H, NH_2_), 7.81–7.86 (m, 2H, C_6_H_4_), 7.95–8.00 (m, 2H, C_6_H_4_), 9.69 (s, 1H, NH); ^13^C NMR (75 MHz, DMSO*-d_6_*/CCl_4_, 1/3): *δ* 18.15, 25.64, 26.42, 28.38, 44.73, 49.51, 52.02, 53.46, 105.00, 118.58, 122.83, 128.47, 130.77, 139.09, 141.97, 143.71, 156.43, 159.30, 160.20, 194.62. Anal. calcd. for C_28_H_35_N_5_O_3_: C 68.69; H 7.21; N 14.30%. Found: C 69.04; H 7.39; N 14.56%. ESI HRMS [C_28_H_35_O_3_N_5_+H^+^] Calculated: 490.2818. Found: 490.2819.

### 3.8. General Procedure for the Synthesis of Compounds ***11a–c***

 ***A.***The same method used for the preparation of compounds **10a–l**. ***B.***A mixture of compound **2** (1 mmol) and the corresponding amine (5 mmol) was refluxed for 30 min. The reaction mixture was cooled, water (50 mL) was added, and the separated crystals were filtered off, washed with water, dried, and recrystallized from ethanol.

3-(Benzylamino)-7-isopropyl-1-pyrrolidin-1-yl-5,6,7,8-tetrahydro-2,7-naphthyridine-4-carbonitrile (**11a**). Yellow solid; yield 72(A)/78(B)%, mp 168–170 °C; IR *ν*/cm^–1^: 3348 (NH), 2186 (C≡N). ^1^H NMR (300 MHz, DMSO*-d_6_*/CCl_4_, 1/3): *δ* 1.06 (d, *J =* 6.5 Hz, 6H, CH(CH_3_)_2_), 1.83–1.89 (m, 4H, C_5_H_8_N), 2.62–2.68 (m, 2H, NCH_2_CH_2_), 2.70–2.76 (m, 2H, NCH_2_CH_2_), 2.80 (sp, *J =* 6.4 Hz, 1H, CH(CH_3_)_2_), 3.43–3.51 (m, 4H, N(CH_2_)_2_), 3.44 (s, 2H, NCH_2_), 4.54 (d, *J =* 6.0 Hz, 2H, NHCH_2_), 6.52 (t, *J =* 6.0 Hz, 1H, NH), 7.10–7.31 (m, 5H, Ph); ^13^C NMR (75 MHz, DMSO*-d_6_*/CCl_4_, 1/3): *δ* 18.13, 24.91, 28.95, 43.88, 44.40, 48.74, 49.03, 53.36, 77.81, 105.84, 117.33, 125.68, 126.73, 127.36, 140.68, 147.87, 156.0, 157.43. Anal. calcd. for C_23_H_29_N_5_: C 73.57; H 7.78; N 18.65%. Found: C 73.89; H 7.94; N 18.89%. ESI HRMS [C_23_H_29_N_5_+H^+^] Calculated: 376.2501. Found: 376.2503.

1-Azepan-1-yl-3-(benzylamino)-7-isopropyl-5,6,7,8-tetrahydro-2,7-naphthyridine-4-carbonitrile (**11b**). Yellow solid; yield 71(A)/76(B)%, mp 127–129 °C; IR *ν*/cm^–1^: 3215 (NH), 2189 (C≡N). ^1^H NMR (300 MHz, DMSO*-d_6_*/CCl_4_, 1/3): *δ* 1.06 (d, *J =* 6.5 Hz, 6H, CH(CH_3_)_2_), 1.47–1.55 (m, 4H, C_6_H_12_N), 1.64–1.72 (m, 4H, C_6_H_12_N), 2.66 (t, *J* = 5.9 Hz, 2H, NCH_2_CH_2_), 2.72–2.84 (m, 3H, NCH_2_CH_2_, CH(CH_3_)_2_), 3.30 (s, 2H, NCH_2_), 3.38–3.43 (m, 4H, N(CH_2_)_2_), 4.55 (d, *J =* 5.9 Hz, 2H, CH_2_Ph), 6.60 (t, *J* = 5.9 Hz, 1H, NH), 7.09–7.28 (m, 5H, Ph); ^13^C NMR (75 MHz, DMSO*-d_6_*/CCl_4_, 1/3): *δ* 18.20, 26.20, 28.20, 29.15, 43.92, 44.66, 49.57, 50.64, 53.34, 79.13, 107.00, 117.09, 125.62, 126.41, 127.37, 140.54, 148.63, 155.73, 159.54. Anal. calcd. for C_25_H_33_N_5_: C 74.40; H 8.24; N 17.35%. Found: C 74.78; H 8.45; N 17.62%. ESI HRMS [C_25_H_33_N_5_+H^+^] Calculated: 404.2814. Found: 404.2816.

1-Azepan-1-yl-3-[(2-furylmethyl)amino]-7-isopropyl-5,6,7,8-tetrahydro-2,7-naphthyridine-4-carbonitrile (**11c**). Yellow solid; yield 74(A)/79(B)%, mp 113–115 °C; IR *ν*/cm^–1^: 3394 (NH), 2199 (C≡N). ^1^H NMR (300 MHz, DMSO*-d_6_*/CCl_4_, 1/3): *δ* 1.07 (d, *J =* 6.5 Hz, 6H, CH(CH_3_)_2_), 1.52–1.60 (m, 4H, C_6_H_12_N), 1.71–1.80 (m, 4H, C_6_H_12_N), 2.66 (t, *J* = 5.6 Hz, 2H, NCH_2_CH_2_), 2.72–2.85 (m, 3H, NCH_2_CH_2_, CH(CH_3_)_2_), 3.31 (s, 2H, NCH_2_), 3.46–3.52 (m, 4H, N(CH_2_)_2_), 4.52 (d, *J =* 5.8 Hz, 2H, NHCH_2_), 6.09 (dd, *J =* 3.2, 0.9 Hz, 1H, 4-CH_fur_), 6.24 (dd, *J =* 3.2, 1.8 Hz, 1H, 3-CH_fur_), 6.40 (t, *J* = 5.8 Hz, 1H, NH), 7.31 (dd, *J =* 1.8, 0.9 Hz, 1H, 5-CH_fur_); ^13^C NMR (75 MHz, DMSO*-d_6_*/CCl_4_, 1/3): *δ* 18.19, 26.26, 28.23, 29.11, 37.40, 44.64, 49.52, 50.69, 53.34, 79.43, 105.33, 107.29, 109.60, 116.90, 140.35, 148.63, 153.57, 155.42, 159.51. Anal. calcd. for C_23_H_31_N_5_O: C 70.20; H 7.94; N 17.80%. Found: C 70.56; H 8.13; N 18.05%. ESI HRMS [C_23_H_31_O_1_N_5_+H^+^] Calculated: 394.2607. Found: 394.2609.

## 4. Conclusions

In summary, an efficient method for the synthesis of new heterocyclic systems, furo[2,3-*c*]-2,7-naphthyridines **6,** and a new method for the synthesis of 1,3-diamino-2,7-naphthyridines **11** were described. The studies have shown that 1-amino-3-[(2-hydroxyethyl)thio]-2,7-naphthyridines **3**, obtained from the alkylation of 1-amino-3-chloro-2,7-naphthyridines **2**, under the action of sodium hydroxide, could undergo Smiles rearrangement with the formation of 1-amino-3-oxo-2,7-naphthyridines **4** in high yields.

It was found that the cyclization reaction of alkoxyacetamides **9** could follow two different pathways depending on their structure. Thus, in the cases of alkoxyacetamides containing cyclic or aromatic amine fragments only the expected aminoamides of furo[2,3-*c*]-2,7-naphthyridine **10** were obtained. In the cases of amides deriving from non-aromatic primary amines the Smiles type rearrangement occurred with formation of 1,3-diamino-2,7-naphthyridines **11**.

## Data Availability

Not applicable.

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
