# Peer review of "Synthesis of 1-Amino-3-oxo-2,7-naphthyridines via Smiles Rearrangement: A New Approach in the Field of Chemistry of Heterocyclic Compounds"

_ijms, 2022, doi:10.3390/ijms23115904_

Round 1

Reviewer 1 Report

The authors described the synthesis of 1-amino-3-oxo-2,7-naphthyridines for the first time, which is highly relevant to be publish. However, the introduction is weak focusing only on the authors previous work, that should be improved namely the sentence about the importance of this heterocycles should be extended. In the sentence " 2,7-Naphthyridine derivatives (...) also because", the authors said "also because" but is only specified one reason.

In the scheme 1, 2 and 3 is not indicated a-b near to the numeration of the compounds, this should be corrected and introduced more clearly according to the scheme 4 .

The scheme 3 is also not very clear, I suggest that the authors should draw the structures 2 and 4 instead of add only the numbers, because if not the X=O, X=S on the arrow is confusing.

Reviewer 2 Report

In the paper, Authors presented a synthesis of 1-amino-3-oxo-2,7-naphthyridines via Smiles rearrangement. I recommend accepting the manuscript for publication after major revision. Paper should be improved.

Manuscript need to be re-written, the flow of paper is poor. Would you explicitly specify the novelty of your work? What progress against the most recent state-of-the-art similar studies was made? Please clearly define the purpose of the study.

Please avoid lumping references as in [1–10] and all other. Instead summarize the main contribution of each referenced paper in a separate sentence.

Please present the "Results and Discussion" part in more detail. Refine table 1 - it is not known where the substituent junctions are, and where the methyl groups are).

Please prepare supplementary information and put the preparation recipes and spectral data there together with SPECTRUMS.

I do not see any conclusions.

These are just some tips. The manuscript should be improved.

Reviewer 3 Report

In this manuscript the authors report a method for the synthesis of new heterocyclic systems using a Smiles rearrangement. However, this reviewer does not consider that the synthesis of this new family of heterocycles has the importance and impact enough to be published in this top journal. Maybe evaluating the biological properties of the new compounds and with an additional mechanism study, the impact of the research could be improved. But at this stage the article would be suitable only for journals such as Synthesis or Tetrahedron.

Anyway, in whatever journal a supporting information with the spectra of all new compounds and perfectly characterized should be mandatory.

Reviewer 4 Report

This work by Sirakanya and colleagues describes the preparation of naphthyridines-based compounds through Smiles-type rearrangement. 

Although the work is an interesting piece of research, there are some points that the authors should address in any subsequent revision:

  1. The English style should be revised.
  2. I wonder why the authors use elemental analysis in some products and exact mass in other products. I suggest that the authors perform exact mass analyses of all the compounds included in the manuscript.
  3. The authors should indicate how they measured the IR spectra (Nujol, KBr, ATR...).
  4. The equipment employed to measure the melting points should be indicated.
  5. The authors should correct the solvents related to NMR. They should indicate "DMSO-d6" instead of "DMSO". 
  6. The authors should obtain single crystals of one of the derivatives of 10 and another one of 11, and provide the X-ray structures in the manuscript. This would undoubtedly confirm the structure of the products and enrich the quality of the work.
  7. I suggest moving the conclusions section before the experimental part to facilitate reading. Furthermore, the last two sentences ("In both cases the yields were...was confirmed by an alternative synthesis") should be removed, since they do not seem suitable for a conclusions section and ending with the synthesis of 11 seems more appropriate, highlighting the importance of the work. 
  8. The authors must submit the supplementary material that they mentioned in the manuscript. As this document is not available, I have not been able to review the spectroscopic data. 

Round 2

Reviewer 2 Report

  1. Please transfer the spectral characteristic (metrics) to SI.
  2. Table 1- 9p- put of Me group.
  3. Check compound numbering throughout the manuscript.
  4. Review the all manuscript again.

Reviewer 4 Report

After careful review of the answers provided by the authors (attached in the response to other reviewer), the current manuscript, and the NMR spectra included in the Supporting Information, I recommend the rejection of the article.

Although the current work is an interesting piece of research, it needs many improvements regarding the characterization of the compounds. For example, the NMR spectra should be repeated performing a higher number of scans in order to reduce the noise/signal ratio. Furthermore, bidimensional experiments should be performed. The authors also should use a software (for example, MestReNova or TopSpin) to analyze the NMR raw data and provide proper spectra in the supporting information, without any annotation by hand on them.

Additionally, the writing of the article should be revised in depth.

Furthermore, the X-ray structures of a compound 10 and a compound 11 would enrich the quality of the work. If it is impossible to obtain X-ray structures, the study of the biological properties of these compounds would be another input to the work, as recommended by another reviewer.

The authors should take their time to make all the necessary changes and, then, resubmit the article.
